# Understanding and Exploring the Food Preferences of Filipino School-Aged Children Through Free Drawing as a Projective Technique

**DOI:** 10.3390/nu16234035

**Published:** 2024-11-25

**Authors:** Melvin Bernardino, Nicole Kate Diaz Sison, Jeanne Carla Bruce, Claudio Tiribelli, Natalia Rosso

**Affiliations:** 1MASLD UNIT, Fondazione Italiana Fegato, 34149 Trieste, Italy; ctliver@fegato.it; 2Department of Life Sciences, University of Trieste, 34100 Trieste, Italy; 3Philippine Council for Health Research and Development, Department of Science and Technology, Taguig City 1631, Philippines; 4Nutrition and Dietetics Area, Colegio de San Juan de Letran, Intramuros, Manila 1002, Philippines; jeannecarla.bruce@letran.edu.ph; 5Department of Nutrition and Dietetics, College of Education, University of Santo Tomas, Espana Manila, Manila 1015, Philippines; ndsison@ust.edu.ph

**Keywords:** children’s food preference, food knowledge, children’s drawing, projective technique, nutrition education

## Abstract

Background and Objectives: Numerous traditional and innovative approaches have been employed to understand and evaluate children’s food preferences and food and nutrition knowledge, recognizing their essential role in shaping good nutrition. Drawing as a projective technique allows children to express their unconscious thoughts and preferences through visual representation, distinguishing it from other methods by providing an insight into their inner feelings and perceptions that may not be easily articulated through verbal techniques. The main goals of the study are to use drawing as a projective technique to gain insights into children’s food preferences, and to examine the children’s current nutrition knowledge and dietary perceptions. Methods: This study involved school-aged children from four public schools in San Jose City, Nueva Ecija, Philippines, who met the inclusion criteria and provided parental consent and the children’s permission. Data collection included (a) questionnaires to measure food group and food frequency knowledge, the children’s opinions on food healthiness and likability, and (b) a drawing activity as a projective technique. The questionnaire scores and the specific foods on the children’s drawings were entered into an electronic worksheet and analyzed quantitatively. Results: The majority of Filipino school-aged children have a low (50%) to average (43%) level of food group knowledge and an average (62%) to low (32%) level of food frequency knowledge. The children can identify the healthiness of the food, but they express a liking for both healthy and unhealthy options. The children’s drawings showed a low preference for Glow food groups, including fruits and vegetables (47%), as compared to Grow foods (94%), Beverages (89%), and Go foods (85%) groups. “Rice and Egg”, the most paired items, indicated a preference among Filipino children. Gender-based analysis showed girls favored “Ice Cream”, “Bread”, “Apple”, and “Oranges” more than boys, but there were no significant gender differences found in Grow food group preferences. Conclusions: Children’s drawings are an effective, valuable complementary tool for understanding children’s food preferences, displaying the value of creative methods in gaining unique insights. The results highlight specific gaps in knowledge, such as the need for a better understanding of food groups and the importance of fruits and vegetables among children. Addressing these gaps in educational programs could enhance children’s food knowledge and encourage healthier dietary choices. Nutrition education programs might use interactive activities focused on food groups and emphasize the benefits of fruits and vegetables to promote better dietary habits for the improvement of children’s long-term health outcomes.

## 1. Introduction

The World Health Organization (2017) [1] reports that numerous countries worldwide are dealing with the double burden of malnutrition, characterized by the co-existence of undernutrition and overnutrition as nutritional status within the population. The existence of the high prevalence of undernutrition and the increasing prevalence of overweight and obesity has resulted in a more challenging nutrition problem.

One of the most vulnerable groups affected by malnutrition are preschool (1–6 years old) and school-aged (7–12 years old) children. The Philippines, a lower-middle-income developing country, is currently suffering from the double burden of malnutrition. Although there is a decreasing trend of underweight, wasting, and stunting, it remains a public health concern. Furthermore, according to the National Nutrition Survey data from the Department of Science and Technology Food and Nutrition Research Institute (DOST-FNRI), childhood overnutrition has continuously increased, particularly “obesity”, “from 7.3% in 2003 to 10.4% in 2019 [2].

One factor affecting children’s nutritional status is their dietary choices, which are significantly influenced by their food preferences. These preferences play a crucial role in determining the quality of their diet [3]. Childhood is a critical period for developing and establishing dietary preferences, as preferences in this life stage persist into later life. This process occurs in utero and continues throughout breastfeeding, formula feeding, and complementary feeding [4]. While exposure to various food flavors and genetic determinants greatly influences initial food preferences during infancy and early childhood, other factors must be considered. Psychological factors, such as emotional eating, food neophobia, and picky eating; socioeconomic factors including food access, affordability, stress, education, and environmental influences; and cultural factors related to traditional food preferences and family eating patterns all interact in a complex way to shape children’s dietary habits and health outcomes [5].

Eating behaviors developed in early childhood can set the stage for a person’s lifelong eating habits. During these formative years, children establish preferences, eating patterns, and attitudes toward food that can influence their overall health. For instance, early exposure to a balanced diet of fruits, vegetables, and whole grains can promote healthy eating habits that reduce the risk of chronic diseases like obesity, diabetes, and heart disease in adulthood [6,7]. Furthermore, the types of foods consumed during childhood impact physical health and have implications for social and behavioral development [8]. Proper nutrition is linked to cognitive function, mood regulation, and behavioral outcomes. Eating a diet rich in essential nutrients like omega-3 fatty acids, vitamins, and minerals supports brain development, enhances focus, and may reduce antisocial behavior [9,10]. A healthy, balanced diet containing the right proportions of macronutrients—such as complex carbohydrates, proteins, and healthy fats—and micronutrients like vitamins and minerals from natural sources like fruits and vegetables work synergistically to enhance cognitive development in children [11]. In contrast, poor diets, which may be high in sugar, processed foods, and/or unhealthy fats can harm cognitive function, leading to issues like depressive and anxiety symptoms and difficulties in social interactions [12,13]. These early eating behaviors can have long-lasting effects, influencing academic performance, mental health, and the ability to form positive social relationships later in life. Therefore, it is crucial to encourage healthy eating habits from a young age for physical health and individuals’ social and emotional well-being.

Numerous studies have established a connection between good nutrition knowledge and dietary intake, as well as perceptions of food. It is widely believed that understanding the health benefits of specific food items can significantly impact an individual’s attitude towards food and their eating habits. An individual’s food choices and nutritional intake are influenced by their awareness of food and their perception of the importance of balanced meals [14]. Various factors shape a child’s nutrition knowledge, including their teachers’ literacy and parenting practices. Therefore, assessing a child’s nutrition knowledge is essential to understand its impact on dietary choices and preferences.

Researchers have employed various traditional and innovative approaches to evaluate children’s food preferences. New methodologies such as projective techniques have emerged continuously, and one example of this is drawing [15,16]. Drawing as a projective technique is a window for studying children’s food preferences because it allows children to express their perceptions and preferences simply and intuitively. The drawing technique allows the conceptualization of food terms or concepts from the children’s language. The drawing technique allows working with children without needing any prior training and does not provide any burden or stress on the child population.

Children’s drawings have been used in the fields of education and psychology to gain more insight into the social, emotional, physical, and intellectual development of each child. By doing this, drawings can thus provide valuable information on the development of children’s environmental perceptions [17]. Children’s drawings have been combined with other techniques, such as interviews, for a better understanding of children’s representation [18]. Because of the fast-paced nature of society, there have been various attempts to automate the drawing of psychological tests by utilizing deep learning technology to process images. In this way, artificial intelligence (AI) will have the ability to analyze children’s drawings and conduct psychological assessments, making it possible to offer this service online or through smartphones for testing purposes [19]. This technological advancement not only makes psychological testing more convenient but also enhances our ability to use children’s drawings as a powerful tool for understanding their developmental progress and providing solutions for early intervention and support.

In the context of the current study, the drawing technique enables the visualization of food terms or concepts using the consumer’s language, including children. Children`s drawings allow them to explore and communicate their thoughts creatively without any constraints they might feel in a direct questioning environment. This is a unique technique that is especially effective for younger children, who may not yet have the language skills to fully articulate their preferences but can express them visually. Research has shown that school-age children prefer drawing over writing as a communication method, because it is a natural, spontaneous activity that requires no training and enhances engagement and enjoyment [20,21,22]. This method also helps maintain children’s attention during tasks, making it essential for researchers to use approaches suited to their understanding and interests [23,24].

The main goal of the study is to gain a deeper understanding of children’s food preferences through drawing as a projective technique. We aim to explore how this method can help uncover children’s food preferences and contribute to developing effective interventions to improve childhood nutrition. Furthermore, we examine the children’s current nutrition knowledge and dietary perceptions. Understanding these preferences and perceptions using innovative techniques will contribute as the basis for future educational initiatives aimed at promoting healthier eating habits and improving childhood nutrition outcomes.

## 2. Materials and Methods

### 2.1. Participants and Venue of the Study

School-aged children were recruited from public schools affiliated with Sidha, an educational non-profit organization in San Jose City, Nueva Ecija, Philippines. Sidha is a group of youth volunteer leaders who advocate for children’s rights by organizing feeding programs and providing individualized and group tutoring sessions emphasizing literacy skills, particularly in reading and writing.

The study venue, located in Central Luzon—a region in the Philippines that blends urban and rural areas—primarily relies on agriculture, contributing significantly to the country’s rice, vegetable, and fruit production. The sample size was determined by identifying the total number of children enrolled in the Division of San Jose City and entering this into the software for sample size estimation. The sample size was calculated using OpenEpi version 3 at a confidence level of 95%, a margin of error of 5%, a design effect of 1.0, a power set of 80%, and an expected prevalence of 50.0%. An additional 10% was allocated for the occurrence of dropouts. The initial computed sample was 500 school-aged children.

Inclusion, exclusion, and withdrawal criteria were established to safeguard the research process and the participants.

The inclusion criteria were as follows:School-aged children between 7–11 years old;Those currently enrolled in Grades 2, 3, or 4 levels at the participating school;Children and their parents/guardians who were willing to participate by signing informed consent and children assent forms;Those able to understand and communicate in the language used for data collection;Children available for the duration of the study period.

Participants who did not meet the inclusion criteria were excluded from the study. Additional exclusion criteria included:Significant dietary restrictions or allergies that could affect participation in food preference assessments;Any medical condition or developmental disorder that may impact food preferences.

Withdrawal criteria, based on scientific merit and safety concerns, included:Inability or refusal to continue participating in study activities as required;Withdrawal of consent from the parent/guardian or the participant;Psychological distress or discomfort reported by the child during activities related to food preferences.

The study was approved by the Institutional Review Board of the Region II Trauma and Medical Center (R2TMC-IRB) under Protocol No. 2024:017, with approval granted in August 2024.

### 2.2. Procedure

Before data collection, participation was obtained by obtaining written assent from the school-age children, and written consent was given from the parents during the parent–teacher meeting. Parents were informed about the study through a letter and asked to return the signed consent form if they agreed to have their children participate. Parents were instructed to contact the researcher if they had further questions or wished to withdraw their consent for their child’s participation in the study. The consent form includes an information sheet outlining the study’s purpose, participant selection criteria, the voluntary nature of participation, procedures, potential risks and discomforts, benefits, confidentiality measures, plans for sharing research findings, the right to withdraw, and contact information for further inquiries.

The data collection took 1 to 2 h, including answering the different questionnaires and the actual drawing activity. The homeroom teacher and trained volunteers facilitated the data collection. At the start of the questionnaire, sociodemographic questions regarding the children’s age in years, year/grade level, and gender were included. This sociodemographic information can demonstrate how these factors enhance the understanding of children’s food preferences based on the context of the study.

### 2.3. Measures

#### 2.3.1. Food Knowledge of the School-Aged Children

The knowledge of school-age children of food was assessed using a validated questionnaire from the study by Gan, Cunanan, and Castro (2019) [25]. The questionnaire was composed of (1) 15 food items that aimed to measure the knowledge of the students on classifying food according to the basic concept of GO-GROW-GLOW food: GO foods are rich in carbohydrates, GROW foods are high in protein, and GLOW foods include fruits and vegetables, which are sources of vitamins and minerals. The GO, GROW, and GLOW food group concept is an educational approach widely used in the Philippines to teach Filipino children about balanced nutrition. This concept emphasized the essentiality of each food group in providing nutrients for human health. (2) Next, a sixteen-item food questionnaire that quantifies food frequency by assigning them to the “eat more”, “eat some”, or “eat a little” classification. Foods classified under “eat more” are rich in nutrients but low in fat, sugars, and salt. Those in the “eat some” category contain moderate levels of fat, sugar, and salt. Foods in the “eat little” group have minimal nutritional value, but are high in fat, sugar, and salt.

#### 2.3.2. Healthy or Unhealthy vs. Like and Dislike (HULD)

The questionnaire aims to assess children’s opinions on a particular food item, whether they consider it “healthy and I like it” (H/L), “healthy and I don’t like it” (H/DL), “not healthy and I like it” (NH/L), or “not healthy and I don’t like it” (NH/DL). The food items included were based on the top commonly consumed foods from the national survey in the Philippines [26]. Fifteen food items were categorized as healthy and fifteen as unhealthy. The HULD questionnaire undergoes content validation using the Survey Instrument Validation Rating Scale [27,28] performed by three licensed nutritionist-dietitians and a licensed food science and technology major professional.

#### 2.3.3. Children’s Drawing

In each class, the teacher acts as the facilitator of the activity. The children are given a piece of paper with an empty plate drawing. Before the actual drawing activity starts, a scenario is presented to the children. The facilitator does not provide any time limit for the children to complete the activity, allowing them to express their preferences freely.

The teacher prompted the activity with the following statement: “Your family went outside for a walk and allowed you to play with your friend. After a tiring day, you feel hungry. Draw on the empty plate the food that you want to eat. At the bottom of the plate, kindly indicate the name of the food you draw”.

### 2.4. Data Analysis

#### 2.4.1. Food Knowledge

The food knowledge score of the students was categorized based on the established cut-off: a low score of 0–5 points, an average score of 6–10 points, and high score of 11–15 or 11–16 points. The distribution frequency determined the number of students under each category across each grade level. This categorization helps assess the overall food knowledge level among school-aged children, enabling educators to tailor nutrition education accordingly.

The children’s answers on the HULD questionnaire were manually coded. All of the answers were totalized and converted into percentages. These percentages were then used to plot the perceived healthiness of the food on the y-axis and the likeness on the x-axis in a quadrant. Each quadrant had the following corresponding descriptions: Quadrant I, “healthy and I like it” (H/L); Quadrant II, “healthy and I don’t like it” (H/DL); Quadrant III, “not healthy and I don’t like it” (NH/DL) and Quadrant IV “not healthy and I like it” (NH/L). The scores on questionnaires were entered into the electronic worksheet and analyzed quantitatively.

#### 2.4.2. Children’s Drawing

##### Classifying Children’s Drawings According to Food Group

The children’s drawings were manually coded and interpreted, using frequency and percentage, by the first author, a registered nutritionist-dietitian. Food items were listed and classified according to the concept of GO, GROW, and GLOW. As a standard tool guide, we followed the Department of Education Order No. 13, series of 2017, also known as the “Policy and Guidelines on Healthy Food and Beverage Choices in Schools”, as a reference for the food items on the classification [29]. Foods not listed in the table but drawn by the children were identified based on the first, second, and third authors’ expertise.

A chi-square analysis was performed to provide gender-based analysis by comparing the frequency of food items depicted in drawings between school-aged girls and boys. It includes both the observed and expected frequencies for each gender and *p*-values to assess statistical significance. The chi-square test was performed for categorical data because it assesses whether there is a statistically significant association between two categorical variables. In this case, the categorical variables are gender (girls versus boys) and the food items depicted.

##### Co-Occurrence Analysis

To further analyze the data, the researchers perform a co-occurrence analysis to determine the typical food combinations that the children prefer. The co-occurrence matrix was imported into Gephi version 3, an open-source network analysis and visualization software and a comprehensive platform that allows users to explore the intricacies of systems, such as networks and structures. Using this platform analysis, the food items were treated as nodes, and their co-occurrences as edges connecting them. The resulting network allowed for the identification of patterns, highlighting which food items frequently co-occurred in the children’s food drawing. Co-occurrence analysis not only highlights individual food preferences but also uncovers favored food pairings and groupings. This method can reveal the common food combinations preferred by many children, offering valuable insights for designing balanced diets.

In co-occurrence analysis, a numerical value of “weight” measured the strength or frequency of the co-occurrence between two food items. “Weight” is a measure to determine the frequency of interaction between two nodes. In a network graph, edge connections that are thick means that they have a high weight, implying that the two nodes interacted a lot with each other in the network.

## 3. Results

### 3.1. Demographic Characteristics of the School-Aged Children

The study included 453 out of 500 children, aged 7 to 11, from four public schools in San Jose City, Nueva Ecija, Philippines. There were seven (7) school-aged children reported to drop out. The study included 225 boys (49.66%) and 228 girls (50.33%). Participants were also evenly distributed across different grade levels: 167 children (37%) from Grade 2 level, 140 children (31%) from Grade 3 level, and 146 children (32%) from Grade 4 level, as shown in Table 1.

According to Piaget’s theory of cognitive development, children between the ages of 7 and 12 typically fall into the concrete operational stage. This stage is characterized by their ability to think logically about concrete events. Children begin to understand concepts such as classification, seriation, and conservation at this stage [30]. As shown in Table 1, the average ages in this study are 7 years for Grade 2, 9 years for Grade 3, and 10 years for Grade 4. Piaget’s theory supports the study’s focus on assessing and enhancing food knowledge, as children in this age range are developmentally ready to comprehend and retain such information.

### 3.2. Knowledge of School-Aged Children on Food

#### 3.2.1. School-Aged Children’s Knowledge of Food Group Classification and Food Frequency

The food knowledge of school-aged children was assessed using a validated questionnaire. In terms of the children’s understanding of food group classification, most of the school-aged children fall into the low score category (50%), followed by the average score category (43%), and only a tiny percentage in the high score category (7%). For food frequency knowledge, most of the students are in the average score category (62%), followed by the low score category (32%), with only a tiny percentage in the high score category (6%). There was no statistically significant difference in the food group classification and food frequency knowledge scores between boys and girls across each grade level. The summary of the score distribution across grade levels is shown in Table 2.

For the food group classification, it was evident that food items classified as “GO” were more straightforward for the students to identify, while they struggled to accurately classify items in the “GROW” and “GLOW” food groups. Regarding food frequency, most school-aged children correctly indicated that water should be consumed more (76%), candy should be eaten a little (59%), rice should be eaten more (57%), and watermelon should be eaten more (54%). Furthermore, many students suggested that pancakes and potato chips should be consumed in moderation (45%). However, many children believed that French fries (44%) and pizza (45%) should be eaten more frequently when these should be consumed less. Similarly, donuts (33%) and burgers (37%) were perceived to be occasional foods, although they should also be consumed less. Lastly, 41% of the children thought that soda drinks should be consumed occasionally, whereas these should be consumed sparingly.

The ranked analysis of food items, based on the number of correct answers for both food group classification and frequency, is summarized in Table 3. The detailed responses from the school-aged children regarding food frequency are presented in Table 4.

#### 3.2.2. Healthy or Unhealthy vs. Like and Dislike (HULD)

The study used a scatter plot to visualize the responses of Filipino school-aged children on the HULD questionnaire. It assesses various food items based on two criteria: how much the food is liked (x-axis), and how healthy the food is perceived (y-axis). As shown in Figure 1a, “healthy food items” such as (1) milk, (2) rice, (3) malunggay/moringa, (9) carrots, (10) mango, (11) corn, (12) boiled sweet potato, (13) apple, (14) nuts, (15) eggplant were classified by the school-aged children as both healthy and liked. Most healthy food items fall in this category, showing a strong positive perception of these foods. Although they belong to the same quadrant, food items such as (5) fish, (4) chicken egg, (6) chicken meat, (8) string beans, (7) pork meat are placed near the center of the graph, indicating a more neutral perception. They are neither strongly liked nor intensely disliked, and their perceived health benefits are moderate. It would be interesting to explore this area further. Moreover, as shown in Figure 1b, “unhealthy food items” such as (A) cake, (B) ice cream, (C) instant noodles, (D) French fries, (E) donuts, (F) soft drinks, (G) flavored juice, (I) chips, (J) bacon, (L) pork chops, (M) ice drops, and (O) fishballs were perceived by the school-aged children as unhealthy, but that they liked the food, while (H) instant coffee, (N) energy drinks, and (K) lollipops were perceived as unhealthy and disliked foods, indicating that they are not as strongly liked as the other unhealthy food items. The scatter plot offers valuable insights into Filipino school-aged children’s food knowledge and food choices.

### 3.3. Interpretation of Children’s Drawing

#### 3.3.1. Categorization of Children’s Food Drawings Based on GO-GROW-GLOW Food Concept

The use of free drawing enables children to depict any food items they choose, unrestricted by specific prompts or guidelines. In the study, children drew 16 food items categorized under the “Go” food group, 16 items under the “Grow” food group, 18 items under the “Glow” food group (vegetables), 12 items in the “Glow” food group (fruits), and 6 items in the “Beverage” category. Refer to Table 5 for a detailed list of specific food items within each food group.

The summary of the inclusion of “Go, Grow, Glow” food groups and beverages in children’s food plate drawings by grade level is shown in Table 6. In the overall analysis, a high percentage of school-aged children included Grow foods (94%) and beverages (89%) as part of their food plate drawing. This was followed by Go foods (85%) and Glow foods (47%) being the least included.

On the usual food-group pattern in children’s drawings, Table 7 presented the representation of food-group combinations across grade levels and genders in children’s plate drawings. It was shown that “GO”, “GROW”, and “BEVERAGE” (38.85%) are the most frequent food-group patterns. It was also observed that this food-group pattern was typical across all grade levels. These results highlight that school-aged children do not usually include Glow foods, such as fruits and vegetables, as part of their food drawings.

Aside from determining the food-group pattern, the study analyzes the frequency of specific food items in children’s food drawings. The most frequently drawn food items by children were: “Rice” in the “GO” food group; “Egg” and “Hotdog” in the “GROW” food group; “Apple” and “Banana” in the “GLOW (fruits)” group; “Eggplant” and “Carrot” in the “GLOW (vegetables)” group; and “Water”, “Milk”, and “Juice drinks” in the beverage category. Examples of children’s drawings are shown in Figure 2.

The chi-square analysis results, detailed in Table 5, revealed that girls exhibit a significantly higher preference for “Ice Cream” and “Bread” (GO food items), as well as “Apple” and “Orange” (GLOW food items), with *p*-values of 0.041, 0.015, 0.023, and 0.008, respectively. Conversely, no significant gender differences were observed for items in the “GROW” food group. These results indicate that, while specific foods have notable gender-based preferences, most items across different food groups do not show substantial gender-related differences in depiction.

#### 3.3.2. Co-Occurrence Analysis of Food Items Present in Children’s Drawing

A co-occurrence analysis was performed in the study to identify which food items, under each food group, usually co-exist in the children’s drawings. In Figure 3, the node (circle) represents food items that appeared in the children’s drawings. The size of the nodes indicates the frequency of specific food items that appear in the children’s food plate drawings. The higher frequency resulted in a more significant node. On the other hand, the lines between the nodes represent the co-occurrence of food items. Strong co-occurrence between the food items (nodes) can be interpreted based on the thickness of the lines. Furthermore, Table 8 shows the overall analysis of food item co-occurrences and their weight. According to the weight analysis, “Rice and Egg” has the highest weight among the combinations, which means that children usually draw them together. This leads to the conclusion that this is the food combination preference of the Filipino children participating in the study. The co-occurrence analysis of food items in children’s plate drawings is shown in Figure 3.

## 4. Discussion

### 4.1. Knowledge of Filipino School-Aged Children on Basic Food Groups and Food Frequency

Fifty percent (50%) of the Filipino school-aged children who participated in the study have a low knowledge of classifying food items, according to the concept of GO, GROW, and GLOW food groups. The findings were consistent with previous research, which underscores a widespread gap in nutritional literacy among children in different countries [31,32,33]. While Filipino school-aged children could quickly identify “GO” foods—likely because these staple food items are in the typical Filipino diet, such as rice and bread [34,35]—they struggled with recognizing “GROW” (protein-rich) and “GLOW” (vitamin- and mineral-rich) foods.

“Grow foods” are nutrient-dense foods rich in protein; examples are meat, poultry, eggs, milk, cheese, beans, and lentils. Protein-rich food helps our body to grow, build, repair muscles and tissue, and strengthen our bones and teeth. Research studies have established that the proper amount of intake of protein is closely related to growth and development; therefore, consuming foods that are rich in protein facilitates catch-up growth in stunted children [36]. In the Philippines, according to the 2019 Expanded National Nutrition Survey (ENNS), stunting remains a moderate public health concern, recording that one (1) in four (4) Filipino school-aged children were stunted (24.9%).

On the other hand, “Glow foods” are nutrient-dense foods rich in micronutrients such as vitamins and minerals, dietary fiber, antioxidants, and phytochemicals, which help regulate bodily processes. All fruits and vegetables belong to this group classification. Numerous studies have established that the consumption of fruits and vegetables is an essential component of a child’s balanced diet, and can reduce the risk of experiencing chronic diseases later in life [37]. In the Philippines, micronutrient deficiency across age groups remains a public health concern. This includes deficiencies in Vitamin A, iodine, zinc, and iron, which can lead to the hidden hungers of malnutrition, such as Vitamin A deficiency, iodine deficiency disorder, zinc deficiency, and iron deficiency anemia [38]. Micronutrient deficiencies can cause both morbidity and mortality. They can lead to serious consequences such as stunting, cognitive impairment, susceptibility to infection, birth defects, and lower school performance [39]. At an early age, Filipinos should correctly recognize Grow and Glow food items because of their significant role in the growth, development, and prevention of deficiency and other diet-related diseases in later life.

Moreover, most Filipino school-aged children who participated in the study have an average to low score on classifying food items in terms of their frequency, indicating a limited understanding of how often different foods should be consumed. This study revealed common misconceptions about consuming food such as French fries, pizza, soda drinks, burgers, and donuts, collectively known as empty calories, low nutrient-dense foods, and highly processed food. Research has revealed that the overconsumption of highly processed food was positively associated with body mass index (BMI), fat mass index, waist circumference, and fasting plasma glucose in children aged 3 to 6 years old, contributing to the development of obesity [40]. Due to the presence of these food items in the market, nutrition education should extend beyond simply identifying food groups. It should also highlight the importance of portion control and limiting the intake of highly processed food.

Understanding this basic concept empowers school-aged children to make informed dietary choices that could be the foundation for a healthy lifestyle throughout their life span. Hence, there is a need to establish a more comprehensive and tailored nutrition education program to increase the understanding of school-aged children of the concept of essential nutrition concerning health. This goes beyond the traditional classroom setup of providing basic nutrition information. Instead, it emphasizes a more hands-on learning experience, through (a) interactive activities like storytelling, where cartoon characters or superheroes make healthy food choices, along with group discussions that use simple language and visuals to explain how balanced nutrition helps them feel strong, grow, and stay active; (b) practical exercises, such as building a balanced plate using cutouts of different foods to learn portion sizes and variety, and hands-on snack preparation of nutritious options; (c) visual tools, including flashcards with common foods on them, which can be used to help children categorize items by food groups; (d) engaging games like bingo or matching activities that feature food items from each group; and (e) rapid technological advancements should be utilized to maximize the use of different media platforms for nutrition education; an intervention study by Gan et al. (2019) [25] proposed the effectiveness of a nutrition game application as a reinforcement intervention to previous nutrition education of school-aged children in the Philippines. The studies showed positive results in terms of nutrition knowledge scores. Research studies highlight the significant contribution of school-based food and nutrition education that develops a positive change in children’s nutrition-related knowledge [31,32,33,41,42].

### 4.2. Perceived Healthiness and Likability of Selected Food Items

Learning about nutrition early on can shape long-term eating habits, helping to lower the risk of chronic diseases and promote overall health. The current study showed that children can identify food items as healthy or unhealthy. However, this understanding does not necessarily influence their preferences or food choices. The Filipino school-aged children defined healthy food items as “perceived as healthy, and they liked it”. Filipino school-aged children have a positive perception of nutritious foods, and there is a possibility that they enjoy eating them. Most of the unhealthy food items were defined by Filipino school-aged children as “perceived as unhealthy, but they liked it”. Despite their ability to identify unhealthy foods, they still like these items. This supports earlier research, which found that even if children know certain foods are unhealthy, it doesn’t stop them from consuming them [43,44]. Our results were parallel with the findings of Varela and Salvador (2014) [45] that children aged 5, 7, and 9 can categorize products as expected regarding their healthiness. The only exception in their study was “nuts”. Interestingly, our results indicate that Filipino school-aged children recognize nuts as a healthy food, offering a different perspective on this specific food item. Further research is needed to explore the relationship between health perception and likability, particularly in the neutral zone for protein-rich food, which school-aged children perceive as moderately healthy and likable.

Even though children may recognize healthy and unhealthy foods, this does not prevent them from liking or consuming them. According to the existing literature, some factors could probably explain this, such as sensory appeal and exposure to digital marketing. Sensory properties play a key role in driving both food preferences and aversion [46]. An appealing presentation, along with a soft and easy-to-eat texture, can strongly enhance the enjoyment of particular foods [47]. Exposure to digital media was found to be associated with sweet, fatty, salty, and bitter taste preferences in children and adolescents [48]. Digital media has become a major platform for various marketing strategies, particularly for food advertisements. Interestingly, children reported a willingness to consume advertised foods based on flavor or taste appeal, regardless of whether they knew if the foods were healthy or not [49]. Despite this, highly processed, unhealthy foods are heavily marketed to children, and their exposure to these advertisements significantly shapes their preferences, tastes, and eating habits [50].

### 4.3. Identification of Children’s Food Preferences Through Drawing as a Projective Technique

#### 4.3.1. Food Group Pattern Shown in Children’s Drawing

The most common food groups found in children’s drawings are categorized as “GO”, “GROW”, and “BEVERAGE”. Notably, these patterns exclude the “GLOW” food group, which includes fruits and vegetables. This is consistent with Rageliene’s (2021) [51] observation that children often omit fruits and vegetables from their meal drawings. However, the current study found that girls were more likely to include fruits and vegetables, particularly “apples and oranges”, reflecting a pattern in other studies, where girls preferred vegetables more. In the literature, studies examining gender differences in food preferences among school-aged children consistently indicated that girls demonstrated a stronger preference for fruits and vegetables compared to boys [52,53,54]. The current study’s findings align with the persistent challenge of children’s fruit and vegetable preferences reflecting low consumption worldwide.

Preference emerges as the strongest mediator of the observed differences in fruit and vegetable consumption between boys and girls [55]. Aside from preference, results from large observational studies unveiled that ability- and opportunity-related factors, such as knowledge, self-efficacy, parental influences, and accessibility, were associated with the likelihood of a daily intake of fruits and vegetables [56]. The link between children’s food preferences and the importance of healthy eating habits is crucial, as early preferences can shape lifelong dietary choices. Children tend to favor foods that they are frequently exposed to and find appealing, often favoring sweet, salty, or highly palatable options. However, establishing healthy eating habits early on—such as a preference for fruits, vegetables, whole grains, and other nutritious foods—can promote better physical and cognitive development, reduce the risk of obesity, and prevent chronic diseases later in life

#### 4.3.2. Co-Occurrence Analysis of Food Shown in Children’s Drawing

At an individual level, food preferences are an essential factor that affects food choices and consumption [57]. The co-occurrence network was analyzed both visually and quantitatively. Strong edges and closely knit clusters indicated popular combinations, while isolated nodes or sparse edges represented infrequent or less favored pairings. The co-occurrence analysis provides a clear view of the food combination usually drawn by school-aged children. The result revealed that Glow foods such as fruits and vegetables do not exist in the most common food combinations. The strong connection between “Rice”, “Egg”, “Hotdog”, and “Water” in the co-occurrence network showed that a balanced diet does not exist in children’s drawings.

In the current study, “Rice” was the most frequently drawn item under the “GO” food category by school-aged children. This reflects not only the Philippines’ agricultural identity, but also the specific location of the study, which is known for its rice production. This finding is aligned with the co-occurrence analysis by Kinoshita et al. (2023) [58], conducted in Fukushima, Japan, among first- and second-graders. The study reported that children who drew more food items were likely to consider a typical set meal, which included rice as a staple food, Hamburg steak as a main dish, and miso soup, a traditional Japanese menu item.

Additionally, in the same study, children often included “cucumber” in their food drawings, as it is a local agricultural product in the region. Both studies highlight how local agricultural products influence children’s food drawings, reflecting their environmental and cultural surroundings. Nutritionists and educators can use this information to develop strategies and programs that create meal options appealing to children’s tastes while promoting nutritional balance. Co-occurrence analysis not only highlights individual food preferences, but also uncovers favored food pairings and groupings.

The results suggest that drawing can be an effective method for adults to evaluate children’s meal preferences and understanding, potentially informing actions to enhance their diets and deepen their awareness of their community study. It suggests that children’s drawings can serve as a valuable complementary tool for adults to understand children’s perceptions of health-related information in everyday life.

#### 4.3.3. Specific Food Items Shown in Children’s Drawings

Most of the Filipino school-aged children who participated in the study predominantly included Grow foods as part of their food-plate drawing. The top five “Grow” food items depicted were chicken eggs, hotdogs, chicken meat, fish, and pork. Aside from those processed foods, hotdogs also appeared frequently in the children’s drawings. In the “Go” food category, rice, pizza, ice cream, loaf bread, cake, and cupcakes were the most commonly drawn items. For beverages, the top items illustrated were water and milk. The frequent appearance of water and milk in children’s drawings indicates that children understand the importance of these beverages as part of a daily diet. However, it is concerning that some sugar-sweetened beverages (SSBs) also appeared in the children’s drawings, such as juice, soda, and chocolate drinks, which are sweetened liquid products that contain various forms of added sugar [59].

The results of children’s drawings are interesting since they reflect the available data about actual dietary consumption in a national nutrition survey in the Philippines. The recent Expanded National Nutrition Survey (ENNS) 2018–2019 revealed that “chicken eggs” are the most commonly consumed food items across all household wealth levels in the Philippines [2]. Urban households consumed more processed meat, with “hotdogs” particularly popular. For school-aged children, rice was the primary energy-giving food, with an average intake of 179 grams daily. While water and milk frequently appeared in children’s drawings, indicating an understanding of their importance, sugar-sweetened beverages (SSBs) like juice, soda, and chocolate drinks were also common. Among Filipino children, sweetened powdered and juice drinks were the most consumed SSBs [60].

The same observation in the study conducted by Goldner, Sosa, and Garitta (2021) [61] identified 29 commonly mentioned food items using both free drawing and free listing methods. Among these, vegetables like tomatoes, carrots, pumpkins, and onions were frequently noted. Their findings align with data from the National Survey of Household Expenses from three different periods—1996–1997, 2004–2005, and 2012–2013. According to the survey, tomatoes were consistently the most widely consumed vegetables. In the same study, they answer the question of whether it is possible to obtain scientific information about food consumption using the children’s drawings. They concluded, given their method of comparison of children’s free drawing and free listing, that children can offer information about food preferences through drawing as a form of communication media.

This highlights the potential of using creative methods like free drawing to study food preferences that might reflect dietary habits and food consumption patterns. Additionally, children’s drawings often represent their perceptions, experiences, daily routines, and food intake. While they are not a replacement for traditional dietary surveys, drawings offer an additional layer of understanding of children’s food preferences and dietary choices, which can be a valuable tool for designing more tailored nutrition education and interventions. The children could express their thoughts and choices about different foods by using drawing as a form of media. This suggests that visual representation can be a valuable tool for understanding children’s preferences, especially when verbal or written communication may be limited. Data from this method can be beneficial information as baselines for developing innovative approaches in nutrition education. We acknowledge some limitations of the study, such as (1) the limited geographic scope that leads to a limited representation of Filipino school-aged children, (2) limited insights on children’s anthropometric measures, behavior, home environment, and underlying factors, (3) the parents’ or guardians’ sociodemographic profile and nutrition knowledge. But, based on our knowledge, this is the first study to combine (a) a traditional type of assessment in questionnaires on knowledge, healthiness perception, and likability, and (b) free drawing as a projective technique to explore the children’s food preferences.

This study opens up future opportunities to provide answers and expand knowledge about the health and nutrition of school-aged children. This is the first study to explore children’s food preferences using free drawing among Filipino school-aged children. Further research should investigate using free drawing as a projective technique alongside traditional food surveys, such as food recall, food diaries, and food frequency questionnaires. Combining free drawing with traditional dietary assessment methods for school-aged children presents several promising research directions: (1) Comparing children’s food drawings with actual food consumption data recorded in food diaries or multiple food recalls could offer an insight into both expressed and underlying food preferences, providing a more comprehensive view of their dietary choices. (2) Future studies might explore how children depict food through drawings, potentially uncovering underlying attitudes toward food or food aversions that are not easily communicated in standard questionnaires. (3) Since drawing enables children to express their thoughts, it can be an effective pre- and post-assessment tool to evaluate changes in food preferences resulting from nutrition education. Integrating free drawing with established techniques could greatly enhance dietary research by capturing children’s unique perspectives on food, which are often shaped by complex, non-verbal influences. This approach could provide unique insights into children’s food preferences and serve as valuable baseline data to develop more personalized and practical nutrition education programs.

There is a need for an in-depth exploration of the relationship between children’s health perceptions and the degree that they like certain foods. The concepts of regional and cultural differences should also be included for a better understanding of how local practices and habits shape perception and likability. Finally, considering the findings from this study, food and nutrition education for school-aged children should be designed with a focus on the critical areas identified. The results highlight specific gaps in knowledge, such as the need for a better understanding of food groups and the importance of fruits and vegetables, which should be addressed in future educational programs. Tailoring these initiatives to target these critical areas could improve children’s dietary choices and overall health outcomes.

## 5. Conclusions

The study provides important insights into Filipino school-aged children’s knowledge of food groups, food frequency, preferences, perceptions of healthiness, and likability. Creative methods, such as children’s drawings, have provided valuable insights into their food preferences, particularly revealing a low preference for fruits and vegetables. For dietitians and health promoters, understanding children’s reluctance towards these food groups can help tailor more engaging and effective interventions to increase fruit and vegetable consumption. In education, this insight can inform curriculum development, encouraging innovative approaches to teaching healthy eating habits. Additionally, this information could be useful for policymakers and researchers aiming to develop strategies to address childhood nutrition and improve public health outcomes.

Nevertheless, the study acknowledges that children’s drawings can serve as a valuable complementary tool in dietary assessment, providing unique insights into their food preferences and perceptions that may not be fully captured through traditional methods. The findings of the study could help support and contribute to the concept that children’s drawings can offer visual insights into their nutritional habits and food preferences and potentially highlight challenges related to food access in various regions of the country. These insights could ultimately inform more targeted and effective public health strategies, exploring children’s food preferences through drawing creates age-appropriate and creative methods. The study provides valuable insights that can inform the development of nutrition education programs aimed at addressing malnutrition. This could be achieved by incorporating characters or foods from their drawings to make campaigns relatable creating visually engaging materials for nutrition education campaigns in the public health setting.

## Figures and Tables

**Figure 1 nutrients-16-04035-f001:**
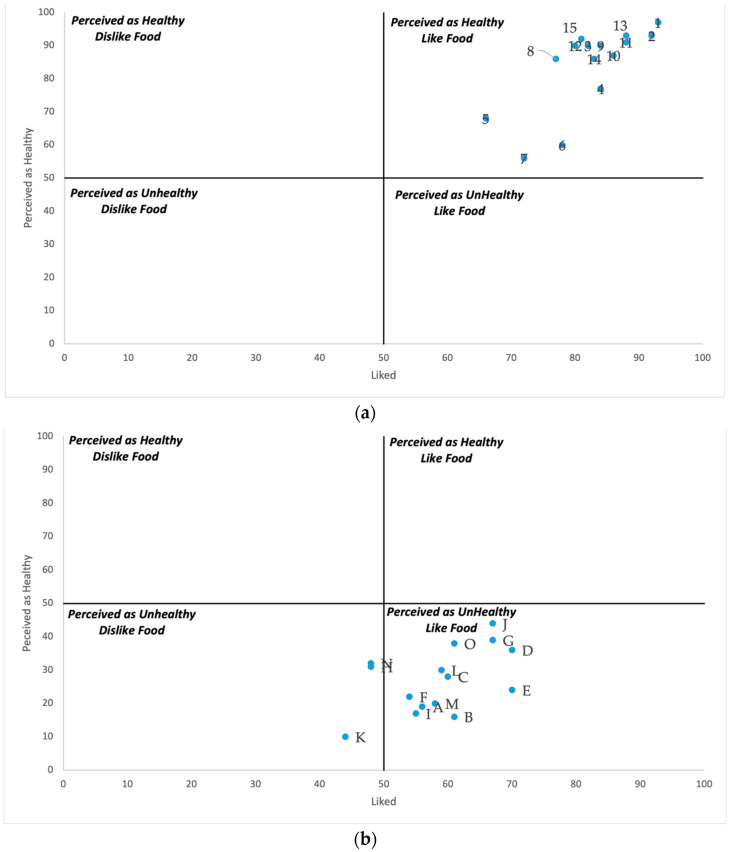
(**a**) Allocation graph for healthy food products on four pre-selected groups. (1) milk; (2) rice; (3) malunggay/moringa; (4) chicken egg; (5) fish; (6) chicken meat; (7) pork meat; (8) string beans; (9) carrots; (10) mango; (11) corn; (12) boiled sweet potato; (13) apple; (14) nuts; (15) eggplant. (**b**) Allocation graph for unhealthy food products on four pre-selected groups. (A) cake; (B) ice cream; (C) instant noodles; (D) French fries; (E) donuts; (F) soft drinks; (G) flavored juice; (H) instant coffee; (I) chips; (J) bacon; (K) lollipops; (L) pork chops; (M) ice drops; (N) energy drinks; (O) fishballs.

**Figure 2 nutrients-16-04035-f002:**
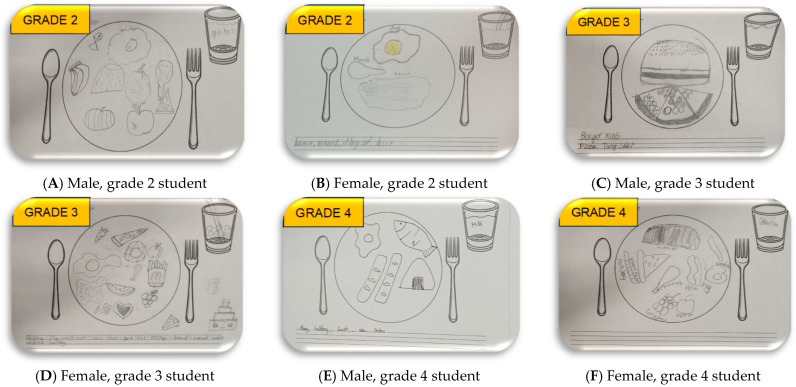
Sample of children’s food plate drawings. (**A**) banana, apple, pizza, chicken, bread, egg, fish, and milk; (**B**) banana, chicken, rice, egg, and fruit juice (**C**) burger, pizza, and water (**D**) egg, carrots, pizza, donut, fries, chicken, watermelon, grapes, apple, cake, rice, (hotdog) sausage, and fruit juice (**E**) rice, (hotdog) sausage, egg, fish, and milk (**F**) rice, (hotdog) sausage, chicken, ham, grapes, apple, watermelon, and milk.

**Figure 3 nutrients-16-04035-f003:**
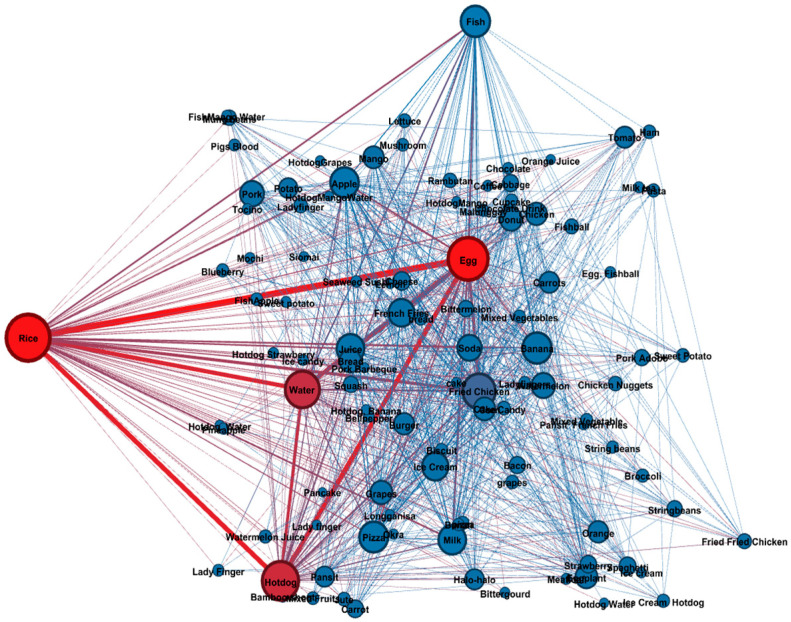
Co-occurrence analysis of food items in children’s plate drawings.

**Table 1 nutrients-16-04035-t001:** Demographics of school-aged children participants by grade level and gender.

School Level	Boys	Girls	Total
Grade 2 (7 years old) *	77	90	167 (37%)
Grade 3 (9 years old) *	66	74	140 (31%)
Grade 4 (10 years old) *	82	64	146 (32%)
Total	225 (49.66%)	228 (50.33%)	453 (100%)

* average age for both boys and girls.

**Table 2 nutrients-16-04035-t002:** Summary of score distribution for food group classification and frequency across grade levels.

*Food Group Classification Knowledge Score*					
Category	Cut off Score	Grade Level II N (%)	Grade Level III N (%)	Grade Level IV N (%)	Total N (%)
Low Score	0–5 points	78 (47)	73 (52)	77 (53)	228 (50)
Average Score	6–10 points	70 (42)	58 (42)	66 (45)	194 (43)
High Score	11–15 points	19 (11)	9 (6)	3 (2)	31 (7)
** *Food Frequency Knowledge Score* **					
**Category**	**Cut off Score**	**Grade Level II** **N (%)**	**Grade Level III N (%)**	**Grade Level IV N (%)**	**Total** **N (%)**
Low Score	0–5 points	50 (30)	51 (36)	44 (30)	145 (32)
Average Score	6–10 points	103 (62)	82 (59)	96 (66)	281 (62)
High Score	11–16 points	14 (8)	7 (5)	6 (4)	27 (6)

**Table 3 nutrients-16-04035-t003:** Ranked analysis of food items by number of correct answers in food classification and frequency.

Food Group Classification				Food Frequency Classification			
Food Item	Correct Food Group	Correct Answers		Food Item	Correct Frequency	Correct Answers	
		N	%			N	%
Rice	Go Food	275	61	Water	Drink More	343	76
Loaf bread	Go Food	266	59	Candy	Eat a Little	269	59
Pandesal	Go Food	259	57	Rice	Eat More	258	57
Pansit	Go Food	215	47	Watermelon	Eat More	246	54
Fish	Grow Food	191	42	Pancake	Eat Some	206	45
Beef	Grow Food	170	37	Potato Chips	Eat a Little	203	45
Carrot	Glow Food	168	37	Hotdog (Sausage)	Eat Some	194	43
Chicken	Grow Food	160	35	Mango	Eat More	186	41
Cheese	Grow Food	155	34	Soda	Drink Some	178	39
Eggplant	Glow Food	151	33	Banana Cue	Eat Some	170	38
Egg	Grow Food	146	32	Burger	Eat Some	165	36
Tomato	Glow Food	141	31	Tomato	Eat More	156	34
Apple	Glow Food	140	31	Pizza	Eat Some	152	34
Milk	Grow Food	129	28	Donut	Eat Some	115	25
Banana	Glow Food	111	24	Ice Cream	Eat Some	92	20
				French Fries	Eat Some	92	20

**Table 4 nutrients-16-04035-t004:** Summary of the responses of school-aged children to food frequency classification.

Food Products	Eat More		Eat Some		Eat a Little	
	N	%	N	%	N	%
Ice cream	88	19	273	60	92	20
Mango	186	41	194	43	73	16
Potato chips	122	27	128	28	203	45
Rice	258	57	163	36	32	7
Pancake	148	33	206	45	99	22
Donut	151	33	187	41	115	25
French fries	199	44	162	36	92	20
Tomato	156	34	149	33	148	33
Burger	166	37	164	36	123	27
Hotdog (sausage)	173	38	194	43	86	19
Watermelon	246	54	143	32	64	14
Banana cue	166	37	170	38	117	26
Pizza	205	45	152	34	96	21
Candy	72	16	112	25	269	59
**Beverages**	**Drink More**		**Drink Some**		**Drink a little**	
	**N**	**%**	**N**	**%**	**N**	**%**
Water	343	76	92	20	18	4
Soda drinks	91	20	184	41	178	39

**Table 5 nutrients-16-04035-t005:** Frequency and chi-square analysis of food items included in children’s drawings by gender.

*Go Food*	Total (N)	%	Girl (N)	%	Expected	Boy (N)	%	Expected	*p*-Value
Rice	291	53.2	148	49.7	146.46	143	57.4	144.54	0.857
Pizza	69	12.6	38	12.8	34.73	31	12.4	34.27	0.431
Ice cream	39	7.1	26	8.7	19.63	13	5.2	19.37	0.041 *
Bread	32	5.9	23	7.7	16.11	9	3.6	15.89	0.015 *
Donut	14	2.6	8	2.7	7.05	6	2.4	6.95	0.610
Noodles	15	2.7	7	2.3	7.55	8	3.2	7.45	0.777
Cake	20	3.7	10	3.4	10.07	10	4.0	9.93	0.976
Halo-halo	8	1.5	6	2.0	4.03	2	0.8	3.97	0.163
Sweet potato	2	0.4	0	0.0	1.01	2	0.8	0.99	0.155
Spaghetti	7	1.3	4	1.3	3.52	3	1.2	3.48	0.719
Potato	42	7.7	22	7.4	21.14	20	8.0	20.86	0.791
Ice candy	3	0.5	2	0.7	1.51	1	0.4	1.49	0.571
Chocolate	1	0.2	1	0.3	0.50	0	0.0	0.50	0.321
Mochi	1	0.2	1	0.3	0.50	0	0.0	0.50	0.321
Biscuit	1	0.2	1	0.3	0.50	0	0.0	0.50	0.321
Cupcake	2	0.4	1	0.3	1.01	1	0.4	0.99	0.993
Total	547		298			249			
** *Grow Food* **	**Total (N)**	**%**	**Girl (N)**	**%**	**Expected**	**Boy (N)**	**%**	**Expected**	***p*-Value**
Hotdog	212	25.3	113	26.5	106.70	99	24.1	105.30	0.387
Chicken	156	18.6	75	17.6	78.52	81	19.8	77.48	0.573
Egg	294	35.1	147	34.5	147.97	147	35.9	146.03	0.910
Pork	33	3.9	16	3.8	16.61	17	4.1	16.39	0.832
Fish	92	11.0	51	12.0	46.30	41	10.0	45.70	0.328
Burger	17	2.0	6	1.4	8.56	11	2.7	8.44	0.215
Chicken nuggets	7	0.8	4	0.9	3.52	3	0.7	3.48	0.719
Fishball	10	1.2	6	1.4	5.03	4	1.0	4.97	0.541
Siomai	1	0.1	1	0.2	0.50	0	0.0	0.50	0.321
Meatloaf	1	0.1	1	0.2	0.50	0	0.0	0.50	0.321
Tocino	2	0.2	1	0.2	1.01	1	0.2	0.99	0.993
Mungbeans	4	0.5	2	0.5	2.01	0	0.0	1.49	0.222
Cheese	4	0.5	1	0.2	2.01	3	0.7	1.99	0.311
Longgannisa	1	0.1	0	0.0	0.50	1	0.2	0.50	0.314
Bacon	3	0.4	1	0.2	1.51	2	0.5	1.49	0.556
Seaweed	1	0.1	1	0.2	0.50	0	0.0	0.50	0.321
Total	838		426			410			
** *Glow foods (Vegetables)* **	**Total (N)**	**%**	**Girl (N)**	**%**	**Expected**	**Boy (N)**	**%**	**Expected**	***p*-Value**
Bamboo shoot	1	0.9	0	0.0	0.50	1	1.6	0.50	0.314
Bell pepper	1	0.9	1	1.9	0.50	0	0.0	0.50	0.321
Bittergourd	5	4.4	1	1.9	2.52	4	6.6	2.48	0.175
Broccoli	6	5.3	3	5.8	3.02	3	4.9	2.98	0.987
Cabbage	8	7.1	5	9.6	4.03	3	4.9	3.97	0.491
Carrot	23	20.4	13	25.0	11.58	10	16.4	11.42	0.553
Corn	2	1.8	1	1.9	1.01	1	1.6	0.99	0.993
Eggplant	24	21.2	10	19.2	12.08	14	23.0	11.92	0.396
Jute	3	2.7	1	1.9	1.51	2	3.3	1.49	0.556
Lady’s finger	9	8.0	4	7.7	4.53	5	8.2	4.47	0.724
Lettuce	2	1.8	0	0.0	1.01	2	3.3	0.99	0.155
Malunggay	1	0.9	0	0.0	0.50	1	1.6	0.50	0.314
Mixed vegetable	11	9.7	4	7.7	5.54	7	11.5	5.46	0.354
Mushroom	3	2.7	2	3.8	1.51	1	1.6	1.49	0.571
Okra	1	0.9	0	0.0	0.50	1	1.6	0.50	0.314
Squash	3	2.7	1	1.9	1.51	2	3.3	1.49	0.556
Stringbeans	5	4.4	3	5.8	2.52	2	3.3	2.48	0.665
Tomato	5	4.4	3	5.8	2.52	2	3.3	2.48	0.665
Total	113		52			61			
** *Glow foods (Fruits)* **	**Total (N)**	**%**	**Girl (N)**	**%**	**Expected**	**Boy (N)**	**%**	**Expected**	***p*-Value**
Apple	74	33.3	47	35.6	37.25	27	30.0	36.75	0.023 *
Banana	51	23.0	22	16.7	25.67	29	32.2	25.33	0.304
Blueberry	1	0.5	1	0.8	0.50	0	0.0	0.50	0.321
Grapes	25	11.3	15	11.4	12.58	10	11.1	12.42	0.334
Lemon	1	0.5	0	0.0	0.50	1	1.1	0.50	0.314
Mango	15	6.8	9	6.8	7.55	6	6.7	7.45	0.454
Mixed fruits	2	0.9	2	1.5	1.01	0	0.0	0.99	0.160
Orange	17	7.7	14	10.6	8.56	3	3.3	8.44	0.008 *
Pineapple	1	0.5	0	0.0	0.50	1	1.1	0.50	0.314
Rambutan	2	0.9	0	0.0	1.01	2	2.2	0.99	0.155
Strawberry	6	2.7	5	3.8	3.02	1	1.1	2.98	0.106
Watermelon	27	12.2	17	12.9	13.59	10	11.1	13.41	0.189
Total	222		132			90			
** *Beverages* **	**Total (N)**	**%**	**Girl (N)**	**%**	**Expected**	**Boy (N)**	**%**	**Expected**	***p*-Value**
Water	204	50.6	115	52.5	102.68	89	48.4	101.32	0.084
Milk	80	19.9	39	17.8	40.26	41	22.3	39.74	0.777
Flavored juice	59	14.6	38	17.4	29.70	21	11.4	29.30	0.031
Soda	50	12.4	21	9.6	25.17	29	15.8	24.83	0.239
Coffee	4	1.0	2	0.9	2.01	2	1.1	1.99	0.989
Chocolate drink	6	1.5	4	1.8	3.02	2	1.1	2.98	0.424
Total	403		219			184			

* Significant at *p*-value < 0.005.

**Table 6 nutrients-16-04035-t006:** Inclusion of ‘Go, Grow, Glow’ food groups and beverages in children’s food plate drawings by grade level.

Food Group	Grade II Level		Grade III Level		Grade IV Level		Total	
	Have Drawn N (%)	Not N (%)	Have Drawn N (%)	Not N (%)	Have Drawn N (%)	Not N (%)	Have Drawn N (%)	Not N (%)
**Go**	144 (86)	23 (14)	113 (81)	27 (19)	126 (86)	20 (14)	383 (85)	70 (15)
**Grow**	157 (94)	10 (6)	132 (94)	8 (6)	139 (95)	7 (5)	428 (94)	25 (6)
**Glow**	74 (44)	93 (56)	67 (48)	73 (52)	70 (48)	76 (52)	211 (47)	242 (53)
**Beverage**	152 (91)	15 (9)	129 (92)	11 (8)	120 (82)	26 (18)	401 (89)	52 (11)

**Table 7 nutrients-16-04035-t007:** Representation of food-group combinations across grades and gender in children’s plate drawings.

Food Group Combinations	Grade II		Grade III		Grade III		Total		
	Girl	Boy	Girl	Boy	Girl	Boy	Girl	Boy	All
“GO only”	1	0	0	0	1	0	2	0	2 (0.44%)
“GROW only”	1	1	0	0	1	0	2	1	3 (0.66%)
“GLOW only”	1	0	1	0	0	1	2	1	3 (0.66%)
“BEVERAGE only “	0	0	0	0	0	0	0	0	0 (0.00%)
“GO” and “GROW”	2	1	3	3	6	2	11	6	17 (3.75%)
“GO” and “GLOW”	2	0	0	0	0	0	2	0	2 (0.44%)
“GO” and “BEVERAGE”	1	0	5	1	1	2	7	3	10 (2.21%)
“GROW” and “GLOW”	3	0	0	2	3	1	6	3	9 (1.99%)
“GROW” and “BEVERAGE”	1	7	11	10	9	4	21	21	42 (9.27%)
“GLOW” and “BEVERAGE”	1	1	0	0	1	1	2	2	4 (0.88%)
“GO”, “GROW”, and “GLOW”	1	4	3	1	3	9	7	14	21 (4.64)
“GO”, “GROW”, and “BEVERAGE”	43	37	23	23	30	20	96	80	176 (38.85)
“GO”, “GLOW”, and “BEVERAGE”	2	1	0	2	4	2	6	5	11 (2.43)
“GROW”, “GLOW”, and “BEVERAGE”	5	4	4	5	2	3	11	12	23 (5.08%)
“GO “GROW”, “GLOW”, and “BEVERAGE”	26	21	24	19	21	19	71	59	130 (28.70%)

**Table 8 nutrients-16-04035-t008:** The overall analysis of food item co-occurrence and their weights.

Food Items with Co-Occurrence	Food Group Combination	Weight
Rice and egg	Go and Grow	201
Rice and hotdog	Go and Grow	153
Hotdog and egg	Grow and Grow	152
Water and egg	Beverage and Grow	136
Water and rice	Beverage and Go	131
Water and hotdog	Beverage and Grow	97
Rice and chicken	Go and Grow	93
Chicken and egg	Grow and Grow	83
Fish and chicken	Grow and Grow	58
Water and chicken	Beverage and Grow	55

## Data Availability

The data presented in this study are available on request from the corresponding author. The data are not publicly available due to ethical restrictions.

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
