# Peer review of "Understanding and Exploring the Food Preferences of Filipino School-Aged Children Through Free Drawing as a Projective Technique"

_nutrients, 2024, doi:10.3390/nu16234035_

Round 1
Reviewer 1 Report
Comments and Suggestions for Authors
It is very interesting study in which authors investigated food preference of Filipino children. And they specially used free drawing method to collect food preference. There are some interesting findings but some technical issues should be addressed further if a revision will be invited as following.
1. It is suggested for authors to say something more about method of network analysis for food relations if possible, which will be helpful for readers understanding related results.
2. It would be better if authors could provide rationality of sample size in the methods.
3. Authors said in the conclusions that “Children's drawings are an effective tool for understanding children’s food preferences…”, but actually it seems not to compare this method in terms of consistency with questionnaire method directly. If possible, authors can compare some selected important food preferences from two kinds of method to find similarity or difference.
4. For table 7, it would be better to show percentage of selection of each food between both genders with Chi-square and P value. Or authors are suggested to say method to estimate P values for smaller sample size.
5. How many foods were found by using free drawing method? Should be reported in the text.
6. A key point should be addressed that baseline information on participants were reported limitedly. Authors just showed information and gender. How about distribution of urban-rural areas?
7. For table 8, what is meaning of “weight”? authors are suggested to clarify it in the context of this study.
Author Response
|
3. Point-by-point response to Comments and Suggestions for Authors |
|
It is very interesting study in which authors investigated food preference of Filipino children. And they specially used free drawing method to collect food preference. There are some interesting findings but some technical issues should be addressed further if a revision will be invited as following.
Comment 1: It is suggested for authors to say something more about method of network analysis for food relations, if possible, which will be helpful for readers understanding related results
Response 1: Thank you for pointing this out. I/We agree with this comment. Therefore we added this text to the manuscript “This finding is aligned with the co-occurrence analysis by Kinoshita et al. (2023) [41], conducted in Fukushima, Japan, among first and second graders. The study reported that children who drew more food items were likely to consider a typical meal set, which included a combination of rice as a staple food, Hamburg steak as a main dish, together with miso soup, a traditional Japanese menu item.” This can be found on Page 16, Line 547-551.
Comment 2: It would be better if authors could provide rationality of sample size in the methods.
Response 2: Thank you for pointing this out. I/We agree with this comment. Therefore we add this text to the manuscript “The sample size was calculated using OpenEpi v.3 at a confidence level of 95%, a margin of error of 5%, a design effect of 1.0, a power set of 80%, and an expected prevalence of 50.8%.10 An additional 20% was allocated for the occurrence of dropouts” This can be found on Page 3-4, Line 150-153.
Comment 3: Authors said in the conclusions that “Children's drawings are an effective tool for understanding children’s food preferences…”, but actually it seems not to compare this method in terms of consistency with questionnaire method directly. If possible, authors can compare some selected important food preferences from two kinds of method to find similarity or difference.
Response 3: We acknowledge the technicality on the use of these words. Thank you for pointing this out. Therefore, we revised this text in the manuscript. “Children's drawings are an effective valuable complementary tool for understanding children's food preferences.” This can be found on Page 1, Line 33-34.
Comment 4: For table 7, it would be better to show percentage of selection of each food between both genders with Chi-square and P value. Or authors are suggested to say method to estimate P values for smaller sample size.
Response 4: Thank you for the suggestion. In the revised version of Table 7, I have included the percentage of selection for each food item by both genders, as recommended. This change makes it easier to compare preferences across genders and provides a clearer view of the data distribution. This can be found on Page 11-13
Comment 5: How many foods were found by using free drawing method? Should be reported in the text.
Thank you for pointing this out. I/We agree with this comment. Therefore we add this text to the manuscript “The use of free drawing enables children to depict any food items they choose, unrestricted by specific prompts or guidelines. In the study, children drew 16 food items categorized under the "Go" food group, 16 items under the "Grow" food group, 18 items under the "Glow" food group (vegetables), 12 items in the "Glow" food group (fruits), and 6 items in the "Beverage" category. Refer to Table 7 for a detailed list of specific food items within each food group”. This can be found on Page 10, Line 368-373.
Comment 6: A key point should be addressed that baseline information on participants were reported limitedly. Authors just showed information and gender. How about distribution of urban-rural areas?
Response 6: Thank you for your feedback. The study site does encompass a mix of both urban and rural areas, which may contribute to a diversity of food experiences and preferences among participants. Because of this detailed demographic information on urban and rural distribution was not included in the baseline characteristics. Future studies could benefit from further stratifying participants by rural, urban and highly urbanized cities/residency to better understand potential variations in food preferences and dietary habits influenced by location.
Comment 7: For table 8, what is meaning of “weight”? authors are suggested to clarify it in the context of this study.
Thank you for pointing this out. I/We agree with this comment. Therefore, we add this text to the manuscript. “In co-occurrence analysis, a numerical value of "weight" measured the strength or frequency of the co-occurrence between two food items. “Weight” is a measure to determine the frequency of interaction between two nodes. In a network graph, edge connections that are thick means that they have a high weight, implying that the two nodes interacted a lot with each other in the network” This can be found on Page 6, Line 274-278.
|

Reviewer 2 Report
Comments and Suggestions for Authors
Interesting study using the drawing technique in children in the Philippines.
This study has important strengths
1. they used validated instruments
2. they had experience in this type of study.
3. they presented interesting statistical analyses, despite the simplicity of the results obtained.
4. the results and discussion were well done
5. the conclusions are consistent with the results and objectives.
Among the weaknesses of the study
1. although the socio-demographic information obtained from the parents is quite poor, more information, educational level, family income, nutritional knowledge of the parents, usual preparations at home, etc. would have provided more information and the information from the parents could have been contrasted with the information from the children.
2. Perhaps some anthropometric measurements such as weight and height would also have enriched the work, perhaps they should be included for future work.
I think that strengths and weaknesses should be mentioned in the manuscript.
Author Response
|
3. Point-by-point response to Comments and Suggestions for Authors |
|
Interesting study using the drawing technique in children in the Philippines..
Comment 1: This study has important strengths • they used validated instruments • they had experience in this type of study. • they presented interesting statistical analyses, despite the simplicity of the results obtained. • the results and discussion were well done • the conclusions are consistent with the results and objectives.
Response: Thank you very much for your positive feedback and thoughtful comments on our study. We are pleased to hear that you recognize the strengths of our approach, including our use of validated instruments, our team's experience with this type of research, and our statistical analyses. We appreciate your remarks on the quality of our results, discussion, and conclusions, as it was our goal to present the findings in a clear and meaningful way. Your encouraging comments are greatly appreciated
Comment 2: Among the weaknesses of the study • Although the socio-demographic information obtained from the parents is quite poor, more information, educational level, family income, nutritional knowledge of the parents, usual preparations at home, etc. would have provided more information and the information from the parents could have been contrasted with the information from the children. • Perhaps some anthropometric measurements such as weight and height would also have enriched the work, perhaps they should be included for future work. I think that strengths and weaknesses should be mentioned in the manuscript.
Response 2: Thank you for pointing this out. I/We agree with this comment. Therefore we add this text to the manuscript” We acknowledge some limitations of the study such as (1) limited geographic scope that leads to the limited representation of Filipino school-aged children (2) limited insights on children`s anthropometric measures, behavior, home environment, and underlying factors (3) parents or guardian sociodemographic profile and nutrition knowledge”. This can be found on Page 17, Line 607-611. This is also revised based on the comment of Reviewer No. 3
|

Reviewer 3 Report
Comments and Suggestions for Authors
This study investigates selected aspect of the dietary pattern and survey and nutritional understanding of Filipino school-aged children through drawing with projective instrument. This demonstrates that most children have poor to moderate understanding of food groups and frequency in a positive manner, choosing unhealthy foods with little frequent sweeps of healthy ones. The study shows that a better understanding of nutrition knowledge is required and that there is a need for education to be directed to children in order to make better food choices.
General Comments:
1. Layout and Formatting:
There are no problems with overall organization of the manuscript and its formatting is clear and logical. However, you must make sure that these figures and tables are labelled appropriately and that titles given give sufficient history for readers to grasp relevance of the particular figure or table without having to go back to the actual text.
2. Use of Pronouns:
Pronouns are used correct most of the time, but there are cases where I think specifying what the subject is would benefit from being clearer. It is advisable not to use too many pronouns, and where there are several authors or participants, to use the names and terms frequently so as not cause confusion.
3. Language and Grammar:
The language is good and formal for the most part; there is however observed an issue with some of the word choices and specificities of some of the sentences in specific. Use correct punctuation, especially in lengthy sentences, to prevent writing compound sentences and improve sense.
4. Linking Findings to Existing Literature:
While the manuscript has cited most of the literature used, there is the need to make these connections more explicit as relates to your results when formulating the discussion section. This will improve the understanding of your findings and demonstrate how they fit into the existing literature.
5. Addressing Limitations:
There are few limitations mentioned, but they could be described in a more detailed manner. Offering a clearer explanation of how these restrictions could have affected the results and the broader transferability of the research would improve the manuscript further.
6. Offering Practical Recommendations for Stakeholders and Future Study:
The recommendations regarding the stakeholders as well as future research are useful, though sometimes generalized. Make specific recommendations or ideas of action or directions for future research would make the study much more valuable. Moreover, stress possible positive outcomes of the application of these recommendations in educational and policy practices.
Abstract:
1. Originality and Value:
As can be seen, the use of drawing as a projective technique in the study has some weaknesses though it was tried and tested as a new method. It is quite possible that depending on children’s drawings, one can inaccurately assess their food choices or even their knowledge. Also, it may not be easy to generalize the results obtained from the sample to the rest of the cultures.
The use of both qualitative and quantitative data provides added depth to the results; however, the qualitative data could be analyzed further to provide for better understanding of the drawings. Last but not least, the findings could be compared to previous studies in order to understand the implications of the study for child nutrition.
2. Revision with Line Number:
· Lines 14-15: Explain the purpose of drawing as a projective technique, and how it differs from other techniques and how it can help a worker understand children’s unconscious preferences.
· Lines 31-32: Enhance the conclusion section by linking the findings to the educational programs and especially supplying examples of the findings.
3. Recommendations:
3.1. You may also want to include the previous method that was used in the abstract, so that readers can have an idea why the present method is innovative.
3.2. Propose that prior to interpreting results it may be beneficial to discuss cultural aspects that can shape meals choices briefly.
3.3 Point out the suggestions for specific educational interventions following the results. For examples, providing the parents and children with short workshops about the necessity of consuming fruits and vegetables.
4. Clarification with Line Number:
4.1. Lines 10-11: Expand on how the drawing as a technique give children a chance to show their preferences apart from speaking.
4.2. Lines 26-27: Explain particular deficiencies detected in the children’s knowledge of food groups and how these could be used to make further improvements to educational endeavors.
5. Recommendations:
Give stronger emphasis on the necessity of helping families headed by females to implement CSOs in the conclusion section and provide very concrete practical recommendations regarding the study findings to the reader.
Keywords:
1. Redundancy:
In this study “children’s food preference” and “children’s drawing” are quite related terms and one may be tempted to think that they refer to the same thing since the children were drawing their choice of food.
2. Specificity:
The keywords are quite descriptive although ‘Children’s food preference’ and ‘Children’s drawing’ are very specific and depict the areas of interest of the study. However, with regards to the postmodern understanding of food literacy “food knowledge” could be more specific. It might be more useful to tie down the sort of knowledge, as for example ‘nutritional knowledge’ or ‘food group knowledge’ to attain better clarity.
3. Extra Keywords:
It could be helpful to include more inclusive topics as well as particular outcomes, terms like “nutrition education,” “projective techniques” or “qualitative studies”. These could help in reaching other people who might be interested in other related topics.
4. Proposed Sequence:
A proposed sequence for the keywords could enhance the logical flow:
1. children’s food preference
2. children’s drawing
3. food knowledge
Such an order of the organization of the study means that the subject of study (children) is presented first, followed by the tools used (drawing), and finally the theme under focus (food knowledge), to make the reader of the study understand the general ideas about the focus of the study on the first glance.
Introduction:
1. Clarity and Flow:
· Lines 39-40: In terms of clarity, I suggest to briefly explain what is meant with ‘double burden of malnutrition’; coexistence of undernutrition and obesity. O
· Lines 45-46: Currently, the shift from focusing on the decline of undernutrition indicators to the growth of obesity can be connected between these two points.
2. Specificity:
· Lines 47-48: Make the obesity statistics time bound and indicate the sources of data used in arriving at the results. Acknowledging where this data comes from will build the argument by giving it more context.
· Lines 55-56: Identify which psychological, socioeconomic, and cultural factors are pertinent as this might offer a broader insight into their impact on diets.
3. Cohesion:
o Lines 58-59: Issues of early eating habits are important in relation to lifelong eating habits. A link between these two concepts can be seen as can be done with more coherence in the following sentence.
4. Evidence Support:
· Lines 66-70: Ideally, the existing literature should be reviewed so that if, for instance, a study is being made to highlight how certain nutrient or food affects the cognitive abilities of a person, then briefly explaining the two most important findings of that study would be a great way of validating the positions been taken.
5. Methodological Discussion:
· Lines 84-85: When introducing new methodologies such as projective mapping and sorting techniques, it will be helpful to describe these briefly and for readers who might not be conversant with either of them.
· Lines 90-92: The advantages of using the drawing technique are well articulated. However, to build an even stronger argument it is recommended to provide a reference to some other authors, who have applied this method, and was successful.
6. Objective Clarity:
· Lines 97-101: The main objective of the study is well stated. But, one more time, it might be useful to define such potential consequences of knowing children’s food preferences as the possible further steps in policy or educational initiatives.
Materials and Methods:
1. Clarity and Structure:
o Lines 104-107: The second element of the brief refers to the description of the venue and which helps to create a better context. Though, the amendments aimed at improving the conciseness and readability: food preferences should specify that the agricultural context is invoked to set the stage for the study.
o Lines 108-119: All regulatory standards concerning admission and exclusion criteria are provided. For better comprehension, it is better to make use of bullets depending on the extent of the criteria which are quite extensive.
2. Specificity:
· Lines 103-104: Provide a short description of the education non-profit organisation so as to give perspective to its application in the research.
· Lines 122-123: The approval from the Institutional Review Board should be followed with the date of approval preferably on a daily basis in research documentation.
3. Procedural Details:
· Lines 125-131: It is understandable how the consent process is done. Nonetheless, you could guarantee the parents were informed of the study’s objective and risks, and where they got this information from.
· Lines 135-135: It is important, however, to explain why questions of sociodemographic characteristics are important and how these characteristics can impact the results of the study.
7. Assessment Tools:
· Lines 137-147: A clear explanation is provided for the theoretical construction of the food knowledge questionnaire. Still, as this section gives an overview of the method, it is useful to name the reasons for selecting the GO-GROW-GLOW framework, though such explanations should be given only briefly as they would belong to a more detailed background section.
· Lines 150-156: The HULD questionnaire is well defined. It is important to specify how the professional validation was conducted (methods which the nutritionist-dietitians have used).
8. Data Analysis:
· Lines 167-171: The scoring categories are still recognizable but, adding a brief note on the importance of such score ranges in terms of everyone’s overall dietary knowledge might prove helpful.
· Lines 179-196: The last Sect, the data analysis section is well done. However, you might consider providing a short description of Gephi for readers who probably have no clue regarding the software as well as including the need for Gephi in the study.
9. Statistical Methods:
· Lines 188-189: State the reason why chi-square analysis has been used to compare categorical data in this context.
· Lines 190-194: When explaining co-occurrence analysis, explain why this method helps in hearing children’s food preference, besides, it helps in identifying those items that may occur together and might be useful for informing diet.
Presentation of Results:
1. Clarity and Structure:
· Line 198-199: The first sentence of the conclusion can be rewritten in a more concise and easily understandable manner.
• The breakdown of the participants by the grade level (Line 202-203) may require further clarification, so that, the reader has the correct perception regarding the percentage.
• The witness of Piaget’s theory (Lines 204-207) seems a bit displaced; better explanation of why this theory helps to coalesce around this study’s conclusion would enrich coherence.
• It would be helpful to make an increased link between the Children’s preference about the kind of foods they want to eat and the significance of eating healthy foods (P280-P 291).
• The labeling of food group classification and frequency knowledge fails to present numeracy (Lines 216-220) that could have enhanced the study.
• The summary tables that are highlighted at Line 240-245 should include some explanation to guide the reader on how the data is arranged. It is possible that the distribution of participants by grade level (Lines 202-203) will require elaboration to make sure the intended percentages are understood by the readers.
• The use of Piaget’s theory in the work (Lines 204-207) seems somewhat clumsy; better connection of this theory to the argument made in the study would promote coherence.
• I think that the relation between children’s preferences concerning the foods and the significance of the educative approaches to healthy eating (280-291) should be emphasized in order to support the significance of the research findings.
• Several ideas in food group classification and knowledge regarding the frequency of its consumption were found to be not tagged with specific numbers (Paragraph IV).
• There should be better explanation as to how the summary tables are laid (Lines 240-245) percentages correctly.
2. Cohesion:
• The reference to Piaget's theory (Lines 204-207) feels somewhat disconnected; a clearer integration of how this theory supports the study's findings would enhance cohesion.
• The link between children's food preferences and the importance of healthy eating habits (Lines 280-291) should be made more explicit to reinforce the relevance of the results.
3. Detail and Specificity:
• The categorization of food group classification and frequency knowledge lacks specific numbers (Lines 216-220), which could provide better context for the findings.
• The organization of the summary tables (Lines 240-245) needs clarification to help readers understand how the data is structured.
4. Data Presentation:
• More ideas could be derived from the interpretation of the scatter plot results (Lines 250-269) especially centered on the importance of positive attitudes towards healthy foods.
• As a suggestion, to enhance the subset of findings related to the analysis of children’s drawings (Lines 279-296), the total number of drawings considered to support the argument should be counted.
• There was confusion in the interpretation of the p-values denoted in the chi-square analysis (Lines 294 – 304) with clearer emphasis needed on what the figures mean for dietary education.
• The definition of “weight” occurs in the co-occurrence analysis (Lines 316-327) should be better placed earlier. Concerning the relative importance of having positive attitudes towards the foods that are good for our health.
• In section with the description of children’s drawings (Lines 279–296) it is suggested to count the number of drawings as well to compare it with the total number of drawings.
• The explanations of the results presented in the chi-square analysis regarding the p-values (Lines 294-304) can be further explained in the context of dietary education.
• In the co–occurrence analysis, the definition of the term “weight” (Lines 316–327) should be provided before this. (Lines 279-296) would benefit from quantifying the total number of drawings analyzed to provide context.
5. Statistical Analysis:
• The significance of p-values in chi-square analysis results (Lines 294-304) could be better articulated in terms of their implications for dietary education.
• The definition of "weight" in the co-occurrence analysis (Lines 316-327) should be included earlier for clarity.
Figures and Tables:
• Make sure all figures and tables referred to in Lines 270-338 are well numbered to enhance understanding.
Discussion
1. Clarity and Cohesion:
· Lines 339-340: The first sentence is rather lengthy would could sound pithier. One way which it would also benefit from is if the authors were more explicit on the results of the study in terms of children’s knowledge.
· Lines 362-365: The recommendation for the implementation of comprehensive nutrition education program could also been more detailed. Prescribing specific techniques or examples might make the notion clearer to readers as well.
2. Repetition:
· Lines 373-376: The information from this section again echoes with repeated mention of the interactive lessons and technology as supportive tools for teaching about nutrition. Some of these could be grouped together to aver unnecessary duplication.
3. Depth of Analysis:
· Lines 384-385: Therefore, even though the study identifies a gap between what children know about their diets and what they actually eat, deepening the reasons why, despite knowing the demerits of the nutrient poor foods, they still prefer to take them could supplement the argument.
· Lines 400-404: The result regarding patterns in the drawings regarding food groups is related and could further discuss the possible rationale for the exclusion of “GLOW” foods that could be discussed.
4. Statistical Evidence:
· Lines 410-412: Issues concerning gender differences in food preferences are covered in the discussion but there are no statistical findings to base the claims on the female’s’ preference for fruits and vegetables on.
5. Link to Existing Literature:
· Lines 427-440: This connection to previous work on food preferences is useful, but could be executed more seamlessly. It is advisable to present more straight-forward comparisons and contrasts between the findings and earlier research.
6. Conclusions and Future Directions:
· Lines 474-481: The limitations of the study are pointed out but could be discussed in more details, especially with regard to the implications of the study limitations in regarding to generalizability of findings.
· Lines 486-490: There is a general lack of focus concerning the possible future research that could be carried out relating to the topic. Such specifications of methodologies or subjects of study would give better ideas for further studies to follow.
7. Structure and Flow:
· Overall: The reader would benefit from proper subheadings to make them distinguish between the sections more easily. It is suggested to use even more different sub-headings each for a sub-part so that students do not have problems with reading them.
Conclusion:
1. Clarity and Conciseness:
• Lines 492-494: The first sentence can be shortened and be made clearer as it is quite long. It might help to divide it into two clear lines just to enhance readability so that there is no misunderstanding of the points being made.
2. Specificity:
• Lines 495-496: The conclusion refers to a “low preference for fruits and vegetables”, it could be clearer how ideas/example/proposal limitation of a low preference would affect/drawer/apply to personnel in dietetics, health promotion, education and who else?
3. Depth of Insight:
• Lines 498-502: To shift the ground further, the section could use such findings to deliberate on how drawings might complement dietary assessment at children level or incorporate it in existing nutrition education program and policy indices.
4. Connection to Study Findings:
• Lines 502-503: The fact that it also stated that some problems associated with the receipt of food are reflected in children’s drawings also appear quite ambiguous. Such has been keeping this state of affairs which by citing from the study could have helped build this case rather than forming this general alarmist belief.
5. Author Contributions:
• Lines 504-506: Hence it’s recommended to reduce the number of words used to enhance the understanding of what is in the text. It could even be highlighted in bullet points just for clarity.
6. Funding Statement:
• Lines 508-511: The format of the funding statement is coherent, but the content might be said in a nut shell. Emphasis on the specific funding method most appropriate for the study and the few most related sources.
7. Overall Impact:
• Lines 512-527: The acknowledgments and other statements that have been made are quite acceptable although there could very slight possibility of existed in more properly organized manner. It might be useful to differentiate the ethical statements (IRB, consent, data availability) into the paragraphs for clearer segregation.
The English could be improved to more clearly express the research.
Author Response
|
3. Point-by-point response to Comments and Suggestions for Authors |
|
GENERAL COMMENTS
Comment 1: Layout and Formatting: There are no problems with overall organization of the manuscript and its formatting is clear and logical. However, you must make sure that these figures and tables are labelled appropriately and that titles given give sufficient history for readers to grasp relevance of the particular figure or table without having to go back to the actual text.
Response 1: Thank you for your valuable feedback on the manuscript. Regarding on your comment on layout and formatting, we have carefully reviewed all figures and tables in the manuscript to ensure they are clearly labeled, with titles that provide sufficient context for standalone comprehension.
Comment 2: Use of Pronouns: Pronouns are used correct most of the time, but there are cases where I think specifying what the subject is would benefit from being clearer. It is advisable not to use too many pronouns, and where there are several authors or participants, to use the names and terms frequently so as not cause confusion.
Response 2: Thank you for this suggestion. In the revised manuscript, we have minimized pronoun usage and consistently used names and specific terms to ensure clarity
Comment 3: Language and Grammar: The language is good and formal for the most part; there is however observed an issue with some of the word choices and specificities of some of the sentences in specific. Use correct punctuation, especially in lengthy sentences, to prevent writing compound sentences and improve sense.
Response 3: Thank you for highlighting this. We have carefully reviewed the language throughout the manuscript, refining word choices and enhancing sentence clarity. Additionally, we have revised the punctuation to avoid overly lengthy sentences and improve readability.
Comment 4: Linking Findings to Existing Literature: While the manuscript has cited most of the literature used, there is the need to make these connections more explicit as relates to your results when formulating the discussion section. This will improve the understanding of your findings and demonstrate how they fit into the existing literature.
Response 4: Thank you for your insightful feedback. In the revised discussion section, we have clarified and strengthened the connections between our findings and the existing literature. We also acknowledge that the literature on children’s drawings related to food preferences is limited, and we have highlighted how our study contributes to this emerging area. We appreciate your guidance on enhancing the clarity and relevance of our discussion.
Comment 5: Addressing Limitations: There are few limitations mentioned, but they could be described in a more detailed manner. Offering a clearer explanation of how these restrictions could have affected the results and the broader transferability of the research would improve the manuscript further.
Response 5: Thank you for this valuable suggestion. We have expanded the limitations section to provide a more detailed discussion of the study’s constraints, including how they may have impacted the results and the broader applicability of our findings.
Comment 6: Offering Practical Recommendations for Stakeholders and Future Study: The recommendations regarding the stakeholders as well as future research are useful, though sometimes generalized. Make specific recommendations or ideas of action or directions for future research would make the study much more valuable. Moreover, stress possible positive outcomes of the application of these recommendations in educational and policy practices.
Response 6: Thank you for your constructive feedback. In response, we have refined the recommendations section to provide more specific actions and directions for future research, along with concrete suggestions for stakeholders. We appreciate your input on how to make this section more impactful.
ABSTRACT
Comment 7: Lines 14-15: Explain the purpose of drawing as a projective technique, and how it differs from other techniques and how it can help a worker understand children’s unconscious preferences.
Response 7: Thank you for pointing this out. We add this text to the manuscript. “ Drawing as a projective technique allows children to express their unconscious thoughts and preferences through visual representation, distinguishing it from other methods by providing insight into their inner feelings and perceptions that may not be easily articulated through verbal techniques” This can be found on Page 1, Line 14-17.
Comment 8: Lines 31-32: Enhance the conclusion section by linking the findings to the educational programs and especially supplying examples of the findings.
Response 8: Thank you for pointing this out. We revised this text to the manuscript by linking the findings to the future possible educational program “Addressing these gaps in educational programs could enhance children's food knowledge and encourage healthier dietary choices. Nutrition education programs might use interactive activities focused on food groups or emphasize the benefits of fruits and vegetables to promote better dietary habits for the improvement of children's long-term health outcomes. This can be found on Page 1, Line 37-40.
Recommendations 1: • You may also want to include the previous method that was used in the abstract, so that readers can have an idea why the present method is innovative. • Propose that prior to interpreting results it may be beneficial to discuss cultural aspects that can shape meals choices briefly. • Point out the suggestions for specific educational interventions following the results. For examples, providing the parents and children with short workshops about the necessity of consuming fruits and vegetables.
Response on Recommendations 1: Thank you for your recommendation, they all are well taken in to the consideration in the manuscript and included in the revision made in Comment 7 and 8.
Clarification with Line Number:
Comment 9: Lines 10-11: Expand on how the drawing as a technique give children a chance to show their preferences apart from speaking.
Response 9: Thank you for pointing this out. To make a unify message we also address this comment together with Comment No. 7. Additionally, to support the Line 10-11 you can find in the introduction Page 3 Line 109-131, supporting discussion on children`s drawing as a projective technique.
To enrich the discussion we added this text to the manuscript: “Children`s drawing allows to explore and communicate their thoughts creatively; without any constraints they might feel in a direct questioning environment. This is a unique technique especially effective for younger children, who may not yet have the language skills to fully articulate their preferences but can express them visually” Page 3 Line 123-127
Comment 10: Lines 26-27: Explain particular deficiencies detected in the children’s knowledge of food groups and how these could be used to make further improvements to educational endeavors.
Response 10: Thank you for pointing this out. To make a unify message we also address this comment together with Comment No. 8.
Recommendation 2: Give stronger emphasis on the necessity of helping families headed by females to implement CSOs in the conclusion section and provide very concrete practical recommendations regarding the study findings to the reader
Response to Recommendation 2: Thank you for your valuable feedback. I appreciate your suggestion to emphasize the necessity of supporting female-headed families and to provide concrete recommendations. However, given that this topic falls outside the primary scope of our study, addressing it in detail might detract from our focus. I’d be happy to consider it as a potential area for further research or for future studies dedicated specifically to this issue
KEYWORDS Comment 11: It could be helpful to include more inclusive topics as well as particular outcomes, terms like “nutrition education,” “projective techniques” or “qualitative studies”. These could help in reaching other people who might be interested in other related topics.
Response 11: The keywords were being revised based on the reviewers comment to observed the logical flow: “children’s food preference, food knowledge, children’s drawing, projective technique, nutrition education” This can be found on Page 1, Line 41-42
INTRODUCTION
Comment 12: Clarity and Flow • Lines 39-40: In terms of clarity, I suggest to briefly explain what is meant with ‘double burden of malnutrition’; coexistence of undernutrition and obesity. • Lines 45-46: Currently, the shift from focusing on the decline of undernutrition indicators to the growth of obesity can be connected between these two points.
Response 12 : To make a clear definition of double burden of malnutrition and to connect these two points. I/we added the text to the manuscript: “The existence of the high prevalence of undernutrition and the increasing prevalence of overweight and obesity has resulted in a more challenging nutrition problem” This can be found on Page 2 Line 48-50
Comment 13: · Lines 47-48: Make the obesity statistics time bound and indicate the sources of data used in arriving at the results. Acknowledging where this data comes from will build the argument by giving it more context.
Response 13: To be more specific on sources of the data, I/we revised the text to the manuscript: “Furthermore, according to the National Nutrition Survey data from the Department of Science and Technology Food and Nutrition Research Institute, childhood overnutrition has continuously increased, particularly "obesity, "from 7.3% in 2003 to 10.4% in 2019” This can be found on Page 2 Line 55-58
Comment 14: · Lines 55-56: Identify which psychological, socioeconomic, and cultural factors are pertinent as this might offer a broader insight into their impact on diets. · Lines 58-59: Issues of early eating habits are important in relation to lifelong eating habits. A link between these two concepts can be seen as can be done with more coherence in the following sentence.
Response 14: To become more specific on the given comments, I/we added the text to the manuscript: “Psychological factors such as emotional eating, food neophobia, and picky eating; socioeconomic factors including food access, affordability, stress, education, and environmental influences; and cultural factors related to traditional food preferences and family eating patterns all interact in a complex way to shape children's dietary habits and health outcomes." This can be found on Page 2 Line 66-70 Additionally, this added text could provide a smooth transition to the next paragraph, which discusses eating behaviors.
Comment 15: Evidence Support:
• Lines 66-70: Ideally, the existing literature should be reviewed so that if, for instance, a study is being made to highlight how certain nutrient or food affects the cognitive abilities of a person, then briefly explaining the two most important findings of that study would be a great way of validating the positions been taken.
Response 15: To provide a support on the claim the text on manuscript has been revised adding existing literature (highlighted in red). “Eating a diet rich in essential nutrients like omega-3 fatty acids, vitamins, and minerals supports brain development, enhances focus, and may reduce antisocial behavior.[9],[10]. A healthy, balanced diet containing the right proportions of macronutrients—such as complex carbohydrates, proteins, and healthy fats—and micronutrients like vitamins and minerals from natural sources like fruits and vegetables works synergistically to enhance cognitive development in children.[11]. In contrast, poor diets such as high in sugar, processed foods, and unhealthy fats can harm cognitive function, leading to issues like depressive and anxiety symptoms and difficulties in social interactions. [12], [13]” This can bee seen on Page 2, Line 81-84
Comment 16: Methodological Discussion: · Lines 84-85: When introducing new methodologies such as projective mapping and sorting techniques, it will be helpful to describe these briefly and for readers who might not be conversant with either of them.
Response 16a: To further describe projective techniques, additional text has been added to the manuscript “New methodologies, such as projective mapping and sorting techniques, have emerged continuously. These user-friendly methods have become increasingly popular in sensory and consumer science. These new methodologies are useful in studying children's food preferences through drawing because they allow children to express their perceptions and preferences in a simple, intuitive way” This can bee seen on Page 3, Line 100-105
• Lines 90-92: The advantages of using the drawing technique are well articulated. However, to build an even stronger argument it is recommended to provide a reference to some other authors, who have applied this method, and was successful.
Response 16b: To strengthen the argument, we provide references that highlight the application of children's drawings in education and psychology, demonstrating their success. Furthermore, we discuss how children's drawings are being studied for potential inclusion in artificial intelligence to enhance services. “Children`s drawing has been used in the field of education and psychology to gained more insight into the social, emotional, physical, and intellectual development of each child. By doing this, drawings can thus provide valuable information on the development of children's environmental perceptions [16]. Children’s drawing has been combined with other techniques such as interviews for better understanding of children representation [17]. Because of the fast-paced society, there have been various attempts to automate the drawing of psychological tests by utilizing deep learning technology to process images. In this way, Artificial intelligence (AI) will have the ability to analyze children's drawings and conduct psychological assessments, making it possible to offer this service online or through smartphones for testing purposes. [18] This technological advancement not only makes psychological testing more convenient but also enhances our ability to use children's drawings as a powerful tool for understanding their developmental progress and providing solutions for early intervention and support.”This can bee seen on Page 3, Line 109-121
Comment 17: Objective Clarity: • Lines 97-101: The main objective of the study is well stated. But, one more time, it might be useful to define such potential consequences of knowing children’s food preferences as the possible further steps in policy or educational initiatives.
Response 17: To define potential consequences of knowing children’s food preferences as the possible further steps in policy or educational initiatives to add this text to the manuscript: “Understanding these preferences and perceptions in the context of the study will contribute as basis for future policy and educational initiatives aimed at promoting healthier eating habits and improving childhood nutrition outcomes with the use of creative techniques applicable to the target population”. This can bee seen on Page 3, Line 137-140
Comment 18: Clarity and Structure: · Lines 104-107: The second element of the brief refers to the description of the venue and which helps to create a better context. Though, the amendments aimed at improving the conciseness and readability: food preferences should specify that the agricultural context is invoked to set the stage for the study.
Response 18 a: To improve the conciseness and readability of the text, the sentence has been paraphrase to highlight that food preferences should specify that the agricultural context is invoked to set the stage for the study. The study venue, located in Central Luzon—a region that blends urban and rural areas—primarily relies on agriculture, contributing significantly to the country's rice, vegetable, and fruit production”. Page 3, Line 148-150
· Lines 108-119: All regulatory standards concerning admission and exclusion criteria are provided. For better comprehension, it is better to make use of bullets depending on the extent of the criteria which are quite extensive.
Response 18 b: Formatted on bullet for better comprehension. Inclusion, exclusion, and withdrawal criteria were established to safeguard the research process and the participants. The inclusion criteria: · school-aged children between 7-11 years old, · those currently enrolled in Grades 2, 3, or 4 at the participating school, · children and their parents/guardians who were willing to participate by signing informed consent and assent forms, · those able to understand and communicate in the language used for data collection, and · children available for the duration of the study period. Participants who did not meet the inclusion criteria were excluded from the study. Additional exclusion criteria included · significant dietary restrictions or allergies that could affect participation in food preference assessments and · any medical condition or developmental disorder that may impact food preferences. Withdrawal criteria, based on scientific merit and safety concerns, included: · inability or refusal to continue participating in study activities as required, · withdrawal of consent by the parent/guardian or the participant, and · psychological distress or discomfort reported by the child during activities related to food preferences. The study was approved by the Institutional Review Board of the Region II Trauma and Medical Center (R2TMC-IRB) under Protocol No. 2024:017, with approval granted in August 2024.
This can be seen on Page 4, Line 155-178
Comment 19: Specificity: · Lines 103-104: Provide a short description of the education non-profit organization so as to give perspective to its application in the research.
Response 19 a: Thank you for pointing this out. I/We agree with this comment. Therefore, we add this text to the manuscript: ” School-aged children were recruited from public schools affiliated with Sidha, an educational non-profit organization in San Jose City, Nueva Ecija, Philippines. Sidha is a group of professional youth volunteer leaders who advocate for children's rights by organizing feeding programs and providing individualized and group tutoring sessions emphasizing literacy skills, particularly in reading and writing”. This can be seen on Page 3, Line 143-147
· Lines 122-123: The approval from the Institutional Review Board should be followed with the date of approval preferably on a daily basis in research documentation.
Response 19b: Thank you for pointing this out. I/We agree with this comment. Therefore, we revised this text to the manuscript: “The study was approved by the Institutional Review Board of the Region II Trauma and Medical Center (R2TMC-IRB) under Protocol No. 2024:017, with approval granted in August 2024”. This can be seen on Page 4, Line 176-178
Comment 20: Procedural Details:
· Lines 125-131: It is understandable how the consent process is done. Nonetheless, you could guarantee the parents were informed of the study’s objective and risks, and where they got this information from.
Response 20a: Thank you for pointing this out. I/We agree with this comment. Therefore, we add this text to the manuscript: “The consent form includes an information sheet outlining the study’s purpose, participant selection criteria, voluntary nature of participation, procedures, potential risks and discomforts, benefits, confidentiality measures, plans for sharing research findings, the right to withdraw, and contact information for further inquiries”. This can be seen on Page 4, Line 185-189.
· Lines 135-135: It is important, however, to explain why questions of sociodemographic characteristics are important and how these characteristics can impact the results of the study.
Response 20b: Thank you for pointing this out. I/We agree with this comment. Therefore, we add this text to the manuscript: “This sociodemographic information can demonstrate how these factors enhance the understanding of children’s food preferences based on the context of the study”. This can be seen on Page 4 Line 193-195
Comment 21: Assessment Tools: · Lines 137-147: A clear explanation is provided for the theoretical construction of the food knowledge questionnaire. Still, as this section gives an overview of the method, it is useful to name the reasons for selecting the GO-GROW-GLOW framework, though such explanations should be given only briefly as they would belong to a more detailed background section.
Response 21a: Thank you for pointing this out. I/We agree with this comment. Therefore, we add this text to the manuscript: “The GO, GROW, and GLOW food group concept is an educational approach widely used in the Philippines to teach Filipino children about balanced nutrition. This concept emphasized the essentiality of each food group in providing nutrients for human health”. This can be seen on Page 5, Line 203-206
· Lines 150-156: The HULD questionnaire is well defined. It is important to specify how the professional validation was conducted (methods which the nutritionist-dietitians have used).
Response 21b: Thank you for pointing this out. I/We agree with this comment. Therefore, we add this text to the manuscript: “The HUDL questionnaire undergoes content validation using the Survey Instrument Validation Rating Scale conducted by three licensed nutritionist-dietitians and a licensed food science and technology major professional” This can be seen on Page 5, Line 218-220
Comment 22: Data Analysis: · Lines 167-171: The scoring categories are still recognizable but, adding a brief note on the importance of such score ranges in terms of everyone’s overall dietary knowledge might prove helpful.
Response 22a: Thank you for pointing this out. I/We agree with this comment. Therefore, we add this text to the manuscript: “This categorization helps to assess the overall food knowledge level among school-aged children, enabling educators to tailor nutrition education accordingly” This can be seen on Page 5, Line 234-236 . · Lines 179-196: The last Sect, the data analysis section is well done. However, you might consider providing a short description of Gephi for readers who probably have no clue regarding the software as well as including the need for Gephi in the study.
Response 22 b: Thank you for pointing this out. I/We agree with this comment. Therefore, we add/revised the text to the manuscript: “The co-occurrence matrix was imported into Gephi, an open-source network analysis, visualization software and a comprehensive platform that allows users to explore the intricacies of complex systems, such as networks and structures”. This can be seen on Page 6, Line 264-267
Comment 23: Statistical Methods: · Lines 188-189: State the reason why chi-square analysis has been used to compare categorical data in this context.
Response 23a: Thank you for pointing this out. I/We agree with this comment. Therefore, we add the text to the manuscript: “The chi-square test was performed for categorical data because it assess whether there is a statistically significant association between two categorical variables. In this case, the categorical variables are gender (girls vs. boys) and the food items depicted”. This can be seen on Page 6, Line 257-260
· Lines 190-194: When explaining co-occurrence analysis, explain why this method helps in hearing children’s food preferences, besides, it helps in identifying food items that may occur together and might be useful for informing diet.
Response 23b: Thank you for pointing this out. I/We agree with this comment. Therefore, we add the text to the manuscript: ”Co-occurrence analysis not only highlights individual food preferences but also uncovers favored food pairings and groupings. This method can reveal common food combinations preferred by many children, offering valuable insights for designing balanced diets. This can be seen on Page 6, Line 270-273
We also, add this text to the manuscript in response the use of the findings in informing the diet via nutrition education efforts
Nutritionists and educators can use this information to develop strategies and programs that create meal options appealing to children’s tastes while promoting nutritional balance. Co-occurrence analysis not only highlights individual food preferences but also uncovers favored food pairings and groupings”. This can be seen on Page 16, Line 554-558
Comment 24: Clarity and Structure: · Line 198-199: The first sentence of the conclusion can be rewritten in a more concise and easily understandable manner.
Response 24a: : Thank you for pointing this out. I/We agree with this comment. Therefore, we revised the text to the manuscript: "The study included 453 children, aged 7 to 11, from four public schools in San Jose City, Nueva Ecija, Philippines." This can be seen on Page 6, Line 281-282
· The breakdown of the participants by the grade level (Line 202-203) may require further clarification, so that, the reader has the correct perception regarding the percentage. · The summary tables that are highlighted at Line 240-245 should include some explanation to guide the reader on how the data is arranged. It is possible that the distribution of participants by grade level (Lines 202-203) will require elaboration to make sure the intended percentages are understood by the readers. · The organization of the summary tables (Lines 240-245) needs clarification to help readers understand how the data is structured.
Response 24b: Thank you for pointing this out. I/We agree with this comment. Therefore, we revised the text to the manuscript: “Participants were also evenly distributed across different grade levels: 167 children (37%) from Grade 2 level, there are 140 children (31%) from Grade 3 level, and 146 children (32%) from Grade 4 level, as shown in Table 1” This can be seen on Page 6, Line 283-285
· The witness of Piaget’s theory (Lines 204-207) seems a bit displaced; better explanation of why this theory helps to coalesce around this study’s conclusion would enrich coherence. · The use of Piaget’s theory in the work (Lines 204-207) seems somewhat clumsy; better connection of this theory to the argument made in the study would promote coherence. · The reference to Piaget's theory (Lines 204-207) feels somewhat disconnected; a clearer integration of how this theory supports the study's findings would enhance cohesion.
Response 24c: Thank you for pointing this out. I/We agree with this comment. Therefore, we revised the text to the manuscript: “The Piaget's theory supports the study's focus on assessing and enhancing food knowledge, as children in this age range are developmentally ready to comprehend and retain such information” This can be seen on Page 6, Line 291-293
· It would be helpful to make an increased link between the Children’s preference about the kind of foods they want to eat and the significance of eating healthy foods (P280-P 291). · The labeling of food group classification and frequency knowledge fails to present numeracy (Lines 216-220) that could have enhanced the study. · The categorization of food group classification and frequency knowledge lacks specific numbers (Lines 216-220), which could provide better context for the findings.
Response 24 d: The summary of score distribution for food group classification and frequency across grade levels are presented in Table 2 and being discussed on Section 3.2.1
· I think that the relation between children’s preferences concerning the foods and the significance of the educative approaches to healthy eating (280-291) should be emphasized in order to support the significance of the research findings.
Resposne 24 e: Thank you for pointing this out. I/We agree with this comment. Therefore, we revised the text to the manuscript: In order to provide a coherent organization, we address this comment on the discussion section.
“This goes beyond the traditional classroom setup of providing basic nutrition information. This goes beyond the traditional classroom setup of providing basic nutrition information. Instead, it emphasizes a more hands-on learning experience through (a) interactive activities like storytelling, where cartoon characters or superheroes make healthy food choices, along with group discussions that use simple language and visuals to explain how balanced nutrition helps them feel strong, grow, and stay active; (b) practical exercises, such as building a balanced plate using cutouts of different foods to learn portion sizes and variety, and hands-on snack preparation of nutritious options; (c) visual tools, including flashcards with common foods, which can be used to help children categorize items by food groups; (d) engaging games like bingo or matching activities that feature food items from each group; and (e) rapid technological advancements should be utilized to maximize the use of different media platforms for nutrition education; an intervention study by Gan et al. (2019) [24] proposed the effectiveness of a nutrition game application as a reinforcement intervention to previous nutrition education of school-aged children in the Philippines” This can be seen on Page 14-15, Line 460-473
Presentation of Results:
Comment 25: Clarity and Structure:
· Several ideas in food group classification and knowledge regarding the frequency of its consumption were found to be not tagged with specific numbers (Paragraph IV).
Response 25a: We have reviewed and ensured that all relevant food group classifications and consumption frequencies are accurately tagged with specific numbers where applicable
· There should be better explanation as to how the summary tables are laid (Lines 240-245) percentages correctly.
Response 25b: Based on our tables we clearly defined the number (N) and its corresponding percentages (%)
Comment 26: Cohesion:
· The link between children's food preferences and the importance of healthy eating habits (Lines 280-291) should be made more explicit to reinforce the relevance of the results.
Response 26: Thank you for pointing this out. I/We agree with this comment. Therefore, we revised the text to the manuscript: In order to provide a coherent organization we address this comment on the discussion section. “The link between children's food preferences and the importance of healthy eating habits is crucial, as early preferences can shape lifelong dietary choices. Children tend to favor foods they are frequently exposed to and find appealing, often favoring sweet, salty, or highly palatable options. However, establishing healthy eating habits early on—such as a preference for fruits, vegetables, whole grains, and other nutritious foods—can promote better physical and cognitive development, reduce the risk of obesity, and prevent chronic diseases later in life”. This can be seen on Page 16, Line 526-532
Comment 27: Data Presentation: · More ideas could be derived from the interpretation of the scatter plot results (Lines 250-269) especially centered on the importance of positive attitudes towards healthy foods. Response 27a: We address this comment in combination with Reviewer 1 Comment 1 point of view. Response: More detailed being discussed on the Discussion part of the paper. This can be seen on Page X, Line X
“This reflects not only the Philippines' agricultural identity but also the specific location of the study, which is known for its rice production. This finding aligns with the co-occurrence analysis by Kinoshita et al. (2023) [48], who observed that children in Fukushima, Japan, among first and second graders, students who drew more food items were likely to draw a typical meal set such as a combination of rice as a staple food, Hamburg steak as a main dish, together with miso soup, a traditional Japanese menu item”. This can be found on Page 16, Line 526-532
· As a suggestion, to enhance the subset of findings related to the analysis of children’s drawings (Lines 279-296), the total number of drawings considered to support the argument should be counted. · In section with the description of children’s drawings (Lines 279–296) it is suggested to count the number of drawings as well to compare it with the total number of drawings.
Response 27b: Thank you for pointing this out. I/We agree with this comment. Therefore, we revised the text to the manuscript: We also take consideration the comment of Reviewer 1 Comment 5 “The use of free drawing enables children to depict any food items they choose, unrestricted by specific prompts or guidelines. In the study, children drew 16 food items categorized under the "Go" food group, 16 items under the "Grow" food group, 18 items under the "Glow" food group (vegetables), 12 items in the "Glow" food group (fruits), and 6 items in the "Beverage" category. Refer to Table 7 for a detailed list of specific food items within each food group” This can be see on Page 10 Line 370-373
· There was confusion in the interpretation of the p-values denoted in the chi-square analysis (Lines 294 – 304) with clearer emphasis needed on what the figures mean for dietary education. · The definition of “weight” occurs in the co-occurrence analysis (Lines 316-327) should be better placed earlier. Concerning the relative importance of having positive attitudes towards the foods that are good for our health. In the co–occurrence analysis, the definition of the term “weight” (Lines 316–327) should be provided before this. (Lines 279-296) would benefit from quantifying the total number of drawings analyzed to provide context. · The definition of "weight" in the co-occurrence analysis (Lines 316-327) should be included earlier for clarity.
Response 27c: Thank you for pointing this out. I/We agree with this comment. Therefore, we revised the text to the manuscript: 2.4.2.2 Co-occurrence analysis To further analyze the data, the researchers perform a co-occurrence analysis to determine the typical food combinations the children prefer. The co-occurrence matrix was imported into Gephi, an open-source network analysis, visualization software and a comprehensive platform that allows users to explore the intricacies of complex systems, such as networks and structures. Using this platform analysis, the food items were treated as nodes, and their co-occurrences as edges connecting them. The resulting network al-lowed for the identification of patterns, highlighting which food items frequently co-occurred in the children's food drawing. In co-occurrence analysis, a numerical value of "weight" measured the strength or frequency of the co-occurrence between two food items. “Weight” is a measure to de-termine the frequency of interaction between two nodes. In a network graph, edge connections that are thick means that they have a high weight, implying that the two nodes interacted a lot with each other in the network. This can be seen on Page 6, Line 263-278
Comment 28 · The explanations of the results presented in the chi-square analysis regarding the p-values (Lines 294-304) can be further explained in the context of dietary education · The explanations of the results presented in the chi-square analysis regarding the p-values (Lines 294-304) can be further explained in the context of dietary education.
Response 28: Girls vs Boys
Comment 29: Figures and Tables: · Make sure all figures and tables referred to in Lines 270-338 are well numbered to enhance understanding.
Response 29: Thank you for the feedback. I already review the figures and tables in that section to ensure they’re correctly numbered for clarity and ease of reference.
DISCUSSION
Comment 30: Clarity and Cohesion: · Lines 339-340: The first sentence is rather lengthy would could sound pithier. One way which it would also benefit from is if the authors were more explicit on the results of the study in terms of children’s knowledge.
Response 30a: Thank you for pointing this out. I/We agree with this comment. Therefore, we revised the text to the manuscript: “Fifty percent (50%) of the Filipino school-aged children who participated in the study have a low knowledge of classifying food items according to the concept of GO, GROW, and GLOW food groups” This can be found on Page 14 Line 435-437.
· Lines 362-365: The recommendation for the implementation of comprehensive nutrition education program could also been more detailed. Prescribing specific techniques or examples might make the notion clearer to readers as well.
Response 30b: Thank you for pointing this out. I/We agree with this comment. Therefore, we revised the text to the manuscript: “This goes beyond the traditional classroom setup of providing basic nutrition information. Instead, it emphasizes a more hands-on learning experience through (a) interactive activities like storytelling, where cartoon characters or superheroes make healthy food choices, along with group discussions that use simple language and visuals to explain how balanced nutrition helps them feel strong, grow, and stay active; (b) practical exercises, such as building a balanced plate using cutouts of different foods to learn portion sizes and variety, and hands-on snack preparation of nutritious options; (c) visual tools, including flashcards with common foods, which can be used to help children categorize items by food groups; and (d) engaging games like bingo or matching activities that feature food items from each group”. This can be seen on Page 14, Line 460-469
Comment 31: Repetition: · Lines 373-376: The information from this section again echoes with repeated mention of the interactive lessons and technology as supportive tools for teaching about nutrition. Some of these could be grouped together to aver unnecessary duplication.
Response 31: Thank you for pointing this out. I/We agree with this comment. Therefore, we revised the text to the manuscript:
Reorganized on the text: (e) rapid technological advancements should be utilized to maximize the use of different media platforms for nutrition education; an intervention study by Gan et al. (2019) [20] proposed the effectiveness of a nutrition game application as a reinforcement intervention to previous nutrition education of school-aged children in the Philippines. The studies showed positive results in terms of nutrition knowledge scores. Research studies highlight the significant contribution of school-based food and nutrition education that develops a positive change in children's nutrition-related knowledge. [23], [24], [25], [29], [30]. This can be seen on Page 14-15, Line 469-476
Comment 32: Depth of Analysis: · Lines 384-385: Therefore, even though the study identifies a gap between what children know about their diets and what they actually eat, deepening the reasons why, despite knowing the demerits of the nutrient poor foods, they still prefer to take them could supplement the argument.
Response 32a: Thank you for pointing this out. I/We agree with this comment. Therefore, we revised the text to the manuscript: “Even though children may recognize healthy and unhealthy foods, it does not prevent them from liking or consuming them. According to the existing literature some factors could probably explain this, such as sensory appeal and exposure to digital marketing. Sensory properties plays a key role in driving both food preferences and aversion.[34]. An appealing presentation, along with a soft and easy-to-eat texture, can strongly enhance the enjoyment of particular foods[35]. Exposure to digital media was found to be associated with sweet, fatty, salty, and bitter taste preferences in children and adolescents [36]. Digital media has become a major platform for various marketing strategies, particularly for food advertisements. Interestingly, children reported a willingness to consume advertised foods based on flavor or taste appeal, regardless of knowing whether the foods were healthy or not [37]. But still highly processed, unhealthy foods that are heavily marketed to children, and exposure to these advertisements sig-nificantly shapes their preferences, tastes, and eating habits.[38]” This can be seen on Page 15, Line 496-508
· Lines 400-404: The result regarding patterns in the drawings regarding food groups is related and could further discuss the possible rationale for the exclusion of “GLOW” foods that could be discussed.
Response 32b: Thank you for pointing this out. I/We agree with this comment. Therefore, we revised the text to the manuscript: “The current study's findings align with the persistent challenge of children's fruit and vegetable preferences reflecting low consumption worldwide. Preference emerges as the strongest mediator of the observed differences in fruit and vegetable consumption between boys and girls[48]. Aside from preference, results from large observational studies unveiled that ability and opportunity-related factors such as knowledge, self-efficacy, parental influences, and accessibility were associated with the likelihood of daily intake of fruits and vegetables[49].” This can be seen on Page 15-16, Line 519-526
Comment 33: Statistical Evidence: · Lines 410-412: Issues concerning gender differences in food preferences are covered in the discussion but there are no statistical findings to base the claims on the female’s’ preference for fruits and vegetables on.
Response 33: We based the claims on the result of the chi-square analysis and we compare it with other studies. You can check this part of the discussion
”This is consistent with Rageliene's (2021) observation that children often omit fruits and vegetables from their meal drawings. However, the current study found that girls were more likely to include fruits and vegetables, particularly "apples and oranges," reflecting a pattern in other studies, where girls preferred vegetables more”. This can be seen on Page 15, Line 513-517
In order to make a stronger we included this text to the manuscript In the literature, studies examining gender differences in food preferences among school-aged children, consistently indicated that girls demonstrated a stronger preference for fruits and vegetables compared to boys [45],[46],[47]. The current study's findings align with the persistent challenge of children's fruit and vegetable preferences reflecting low consumption worldwide. Preference emerges as the strongest mediator of the observed differences in fruit and vegetable consumption between boys and girls[48]. This can be seen on Page 15, Line 517-523
Comment 34: Link to Existing Literature: · Lines 427-440: This connection to previous work on food preferences is useful, but could be executed more seamlessly. It is advisable to present more straight-forward comparisons and contrasts between the findings and earlier research.
Response 34: Thank you for pointing this out. I/We agree with this comment. Therefore, we revised the text to the manuscript: The same observation with the study conducted by Goldner, Sosa, and Garitta (2021) that identified 29 commonly mentioned food items using both free drawing and free listing methods. Among these, vegetables like tomatoes, carrots, pumpkins, and onions were frequently noted. Their findings align with data from the National Survey of Household Expenses from three different periods—1996-1997, 2004-2005, and 2012-2013. According to the survey, tomatoes were consistently the most widely consumed vegetables. In the same study they answer the question if it is possible to obtain scientific information about food consumption using the children`s drawing. They concluded given their method of comparison on children free drawing and free listing; children can offer information about food preferences through drawing as a form of communication media”. This can be seen on Page 15, Line 586-596
CONCLUSIONS AND FUTURE DIRECTIONS:
Comment 35: · Lines 474-481: The limitations of the study are pointed out but could be discussed in more details, especially with regard to the implications of the study limitations in regarding to generalizability of findings.
Response 35a: Thank you for pointing this out. I/We agree with this comment. Therefore, we added the text to the manuscript: “We acknowledge some limitations of the study such as (1) limited geographic scope that leads to the limited representation of Filipino school-aged children (2) limited insights on children`s anthropometric measures, behavior, home environment, and underlying factors (3) parents or guardian sociodemographic profile and nutrition knowledge” This can be seen on Page 17, Line 607-611
Note: We also include the recommendations provided by the Reviewer No 2.
· Lines 486-490: There is a general lack of focus concerning the possible future research that could be carried out relating to the topic. Such specifications of methodologies or subjects of study would give better ideas for further studies to follow.
Response 35b: Thank you for pointing this out. I/We agree with this comment. Therefore, we added the text to the manuscript:
“Combining free drawing with traditional dietary assessment methods for school-aged children presents several promising research directions: (1) Comparing children’s food drawings with actual food consumption data recorded in food diaries or multiple food recalls could offer insight into both expressed and underlying food preferences, providing a more comprehensive view of their dietary choices. (2) Future studies might explore how children depict food through drawings, potentially uncovering underlying attitudes toward food or food aversions that are not easily communicated in standard questionnaires. (3) Since drawing enables children to express their thoughts, it can be an effective pre- and post-assessment tool to evaluate changes in food preferences resulting from nutrition education. Integrating free drawing with established techniques could greatly enhance dietary research by capturing children’s unique perspectives on food, often shaped by complex, non-verbal influences.” This can be seen on Page 17-18, Line 620-631
Comment 36: Structure and Flow: · Overall: The reader would benefit from proper subheadings to make them distinguish between the sections more easily. It is suggested to use even more different sub-headings each for a sub-part so that students do not have problems with reading them.
Response 36 : Thank you for your feedback. I understand the importance of clear subheadings to guide the reader through each section. I'll make sure to add more distinct subheadings for each sub-part to enhance readability and make it easier for students to follow along.
Comment 37 Clarity and Conciseness: · Lines 492-494: The first sentence can be shortened and be made clearer as it is quite long. It might help to divide it into two clear lines just to enhance readability so that there is no misunderstanding of the points being made.
Response 37: Thank you for pointing this out. I/We agree with this comment. Therefore, we revised the text to the manuscript: “The study provides important insights into Filipino school-aged children on their knowledge of food groups, food frequency, preferences, perceptions of healthiness, and likability”. This can be seen on Page 18, Line 645-647
Comment 38: Specificity: · Lines 495-496: The conclusion refers to a “low preference for fruits and vegetables”, it could be clearer how ideas/example/proposal limitation of a low preference would affect/drawer/apply to personnel in dietetics, health promotion, education and who else?
Response: Thank you for pointing this out. I/We agree with this comment. Therefore, we added/revised the text to the manuscript: “For dietitians and health promoters, understanding children’s reluctance towards these food groups can help tailor more engaging and effective interventions to increase fruit and vegetable consumption. In education, this insight can inform curriculum development, encouraging innovative approaches to teaching healthy eating habits. Additionally, this information could be useful for policymakers and researchers aiming to develop strategies to address childhood nutrition and improve public health outcomes." This can be seen on Page 18, Line 649-654
Comment 39: Depth of Insight: · Lines 498-502: To shift the ground further, the section could use such findings to deliberate on how drawings might complement dietary assessment at children level or incorporate it in existing nutrition education program and policy indices.
“Combining free drawing with traditional dietary assessment methods for school-aged children presents several promising research directions: (1) Comparing children’s food drawings with actual food consumption data recorded in food diaries or multiple food recalls could offer insight into both expressed and underlying food preferences, providing a more comprehensive view of their dietary choices. (2) Future studies might explore how children depict food through drawings, potentially uncovering underlying attitudes toward food or food aversions that are not easily communicated in standard questionnaires. (3) Since drawing enables children to express their thoughts, it can be an effective pre- and post-assessment tool to evaluate changes in food preferences resulting from nutrition education. Integrating free drawing with established techniques could greatly enhance dietary research by capturing children’s unique perspectives on food, often shaped by complex, non-verbal influences.” This can be seen on Page 17-18, Line 620-631
Comment 40: Connection to Study Findings: · Lines 502-503: The fact that it also stated that some problems associated with the receipt of food are reflected in children’s drawings also appear quite ambiguous. Such has been keeping this state of affairs which by citing from the study could have helped build this case rather than forming this general alarmist belief.
Response: There are limited studies available exploring the children`s drawing and its application to nutrition. But with thorough search and analysis we make sure that we utilize all of the study available on different journals in order to provide a logical discussion
In summary this are the studies we utilized and included in the different part of the manuscript:
This is consistent with Rageliene's (2021) observation that children often omit fruits and vegetables from their meal drawings. However, the current study found that girls were more likely to include fruits and vegetables, particularly "apples and oranges," reflecting a pattern in other studies, where girls preferred vegetables more.
Reference: T. RagelienÄ—, “Do children favor snacks and dislike vegetables? Exploring children’s food preferences using drawing as a projective technique. A cross-cultural study,” Appetite, vol. 165, p. 105276, Oct. 2021, doi: 10.1016/j.appet.2021.105276.
This finding aligns with the co-occurrence analysis by Kinoshita et al. (2023) [47], who observed that children in Fukushima, Japan, among first and second graders, students who drew more food items were likely to draw a typical meal set such as a combination of rice as a staple food, Hamburg steak as a main dish, together with miso soup, a traditional Japanese menu item. References: L. Kinoshita et al., “Daily meals in context: A quantitative analysis of elementary school students’ drawings,” Front. Commun., vol. 8, Mar. 2023, doi: 10.3389/fcomm.2023.1008108.
The results of children`s drawings are interesting since they reflect the available data about actual dietary consumption data in a national nutrition survey in the Philippines. The same observation with the study conducted by Goldner, Sosa, and Garitta (2021) that identified 29 commonly mentioned food items using both free drawing and free listing methods.
Reference: M. C. Goldner, M. Sosa, and L. Garitta, “Is it possible to obtain food consumption information through children’s drawings? Comparison with the Free Listing,” Appetite, vol. 160, p. 105086, May 2021, doi: 10.1016/j.appet.2020.105086.
Comment 41: Author Contributions
· Lines 504-506: Hence it’s recommended to reduce the number of words used to enhance the understanding of what is in the text. It could even be highlighted in bullet points just for clarity.
Response 41: Thank you for your valuable observation. We appreciate your feedback regarding the author contribution section and funding statement. However, based on our understanding, the format for this section follows the template provided by the Nutrients journal's word template, which outlines how the authors' contributions should be presented. Thank you again for your thoughtful input!
Comment 42: Funding Statement:
· Lines 508-511: The format of the funding statement is coherent, but the content might be said in a nut shell. Emphasis on the specific funding method most appropriate for the study and the few most related sources.
Response 42: Thank you for your comment regarding the funding statement. The authors fully agree with the importance of transparency to avoid any potential conflicts of interest in the future. As such, we have ensured that all funding sources are clearly stated to provide complete transparency regarding any financial support received by the authors mentioned even if it`s not directly benefiting the study. We appreciate your attention to this matter and believe that this approach will help maintain the integrity of the research.
Comment 43: Overall Impact: · Lines 512-527: The acknowledgments and other statements that have been made are quite acceptable although there could very slight possibility of existed in more properly organized manner. It might be useful to differentiate the ethical statements (IRB, consent, data availability) into the paragraphs for clearer segregation.
Response 43: Thank you for your valuable observation. We appreciate your feedback regarding this section. However, based on our understanding, the format for this section follows the template provided by the Nutrients journal's word template.
Thank you for your suggestion. I’ve already made revisions to the text/line in the manuscript to address this. I appreciate your input in improving our paper.
|
|
|

Round 2
Reviewer 1 Report
Comments and Suggestions for Authors
Authors have addressed most of my comments, and the following minor issues are suggested to be addressed further.
1. Athough authors provided sample size estimation in the revised version, it is still unclear about what prevalence was used for this estimation.
2. Authors said they did not collect enough information on basic features of participants. This is likely a limiation of this study, and authors are suggested to address it in the discussion clearly.
Author Response
|
Point-by-point response to Comments and Suggestions for Authors
Comment 1: Although authors provided sample size estimation in the revised version, it is still unclear about what prevalence was used for this estimation. Response 1: At the start, we collected the number of enrolled students from the Division of San Jose City from that data we computed the sample size using the Open Epi Version 3. To further clarify this on the manuscript we added this text to the manuscript: The sample size was determined by identifying the total number of children enrolled in the Division of San Jose City and entering it into the software for sample size estimation. The sample size was calculated using OpenEpi version 3 at a confidence level of 95%, a margin of error of 5%, a design effect of 1.0, a power set of 80%, and an expected prevalence of 50.0%. An additional 10% was allocated for the occurrence of dropouts. The initial computed sample is 500 school-aged children. This can be seen on Page 3, Line 148-153 Comment 2: The authors said they did not collect enough information on basic features of participants. This is likely a limitation of this study, and authors are suggested to address it in the discussion clearly. Response 2: Thank you for pointing this out, we already included the limitations as also recommended by reviewer 2: “We acknowledge some limitations of the study, such as (1) limited geographic scope that leads to the limited representation of Filipino school-aged children, (2) limited insights on children`s anthropometric measures, behavior, home environment, and underlying factors, (3) parents or guardian sociodemographic profile and nutrition knowledge”. |
||
|
This can be seen on Page 3, Line 631-635
|

Reviewer 3 Report
Comments and Suggestions for Authors
Lines 12-14:
Even in the introduction of the presentation, phrases such as “over the years”, which are not only redundant but also unhelpful are used.
Lines 19-22:
Sample size is missing.
Lines 23-24:
There is the language of informal communication (“spreadsheet”).
Line 99-108:
Some concepts, for instance, projective mapping are explained in the introduction yet they appear directionless and unfit for the introduction.
Lines 150-152:
Calculation of sample size included using OpenEpi software and no rationale is given for using an expected prevalence of 50.8%.
Lines 244-253:
Classification of the food items in this study using a policy document originating from the government may enforce bias due to an outside source.
Table 2 and 3: Well compiled, but it would be advisable to add a comment on the readiness threshold levels used to sort the scores as Low, Average and High so as to make it easy for consumers to understand.
Figure 1: The visualization employed in the scatter plot section can be considered successful. However, it would be convenient to draw the trend lines or marks the significant results for comparing the healthy and the unhealthy food items.
Lines 509-511:
The following are the observed deficiencies with reference to the report: Weak transition from results to the generalization of the results. In what ways does the study and the identified patterns connect to other nutritional problems in the Philippines?
Line 634-638:
A great strength is that the conclusion effectively sums up the findings made in the course of the analysis, while the final message is rather weak. Could you insert a more engaging phrase which could serve as an umbrella to the more specific findings of the research in terms of the public health implications?.
The English could be improved to more clearly express the research.
Author Response
|
Point-by-point response to Comments and Suggestions for Authors Reviewer 3 Comment 1: Lines 12-14: Response 1: Thank you for pointing this out, we already paraphrase the sentence to make it more comprehensible and readable: Abstract part: “Numerous traditional and innovative approaches have been employed to understand and evaluate children's food preferences and food and nutrition knowledge, recognizing its essential role in shaping good nutrition” This can be found on Page 1, Line 12-14 Introduction: Researchers employed various traditional and innovative approaches to evaluate children's food preferences. New methodologies such as projective techniques have emerged continuously and one example is drawing [15],[16]. This can be found on Page 3, Line 99-101 Comment 2: Lines 19-22: Response 2: Thank you so much for pointing this out, we already revised the text: The sample size was determined by identifying the total number of children enrolled in the Division of San Jose City and entering it into the software for sample size estimation. The sample size was calculated using OpenEpi version 3 at a confidence level of 95%, a margin of error of 5%, a design effect of 1.0, a power set of 80%, and an expected prevalence of 50.0%. An additional 10% was allocated for the occurrence of dropouts. The initial computed sample is 500 school-aged children. This can be seen on Page 3, Line 148-153 We also specified it in the results section: The study included 453 out of 500 children, aged 7 to 11, from four public schools in San Jose City, Nueva Ecija, Philippines. There were seven (7) school-aged children reported to drop out. This can be seen on Page 6, Line 280-282 Comment 3: Lines 23-24:
Comment 4: Line 99-108: Response 4: Thank you for pointing this out, We already revised the text in the manuscript: See how it is being revised: Old form: Over the years (2000-2015), researchers employed various traditional and innovative approaches to evaluate children's food preferences. New methodologies, such as projective mapping and sorting techniques, have emerged continuously. These user-friendly methods have become increasingly popular in sensory and consumer science. These new methodologies are useful in studying children's food preferences through drawing because they allow children to express their perceptions and preferences in a simple, intuitive way [15]. The drawing technique allows the conceptualization of food terms or concepts from the consumer's language, including children. The drawing technique allows working with children without needing any prior training and does not provide burden and stress on the children population. Revised form: Researchers employed various traditional and innovative approaches to evaluate children's food preferences. New methodologies such as projective techniques have emerged continuously and one example is drawing[15],[16]. Drawing as a projective technique is a window for studying children's food preferences because it allows children to express their perceptions and preferences simply and intuitively. The drawing technique allows the conceptualization of food terms or concepts from the children`s language. The drawing technique allows working with children without needing any prior training and does not provide burden and stress on the children population” This can be found on Page 3, Line 99-106 Comment 5: Lines 150-152: Response 5: The 50.08% expected prevalence is automatically given by the software as automatically input 50.08 if the anticipated % frequency/prevalence is unknown. Comment 6: Lines 244-253: Response 6: Using a government policy document for classifying food items can introduce some degree of bias, as the classification reflects the priorities and criteria set by the policymakers rather than those specific to our study. However, it also ensures alignment with standardized guidelines, facilitating comparability with other studies and policies. To address potential bias, the first, second, and third authors of the study, all of whom are Registered/Licensed Nutritionist-Dietitians, thoroughly reviewed the guidelines and food item classifications. We believe that these guidelines were initially evaluated by the Department of Science and Technology - Food and Nutrition Research Institute (DOST-FNRI), recognized as the nation's reliable authority on nutrition information. We also indicate that “Foods not listed in the table but drawn by the children were identified based on the first, second, and third authors' expertise” This can be seen on Page 6, Line 251-253.
Comment 7: Table 2 and 3: Well compiled, but it would be advisable to add a comment on the readiness threshold levels used to sort the scores as Low, Average and High so as to make it easy for consumers to understand. Response 7: Table 2: Thank you for your comment, we believe that in Table 2, we already include the readiness threshold see the text in red Table 2. Summary of Score Distribution for Food Group Classification and Frequency Across Grade Levels.
Table 3: since this is a ranked analysis, we cannot include the readiness threshold levels as we only intend to analyze the food items that school-aged children typically answered correctly and incorrectly in both food group classification and food frequency assessments Comment 8: Figure 1: The visualization employed in the scatter plot section can be considered successful. However, it would be convenient to draw the trend lines or marks the significant results for comparing the healthy and the unhealthy food items. Response 8: We appreciate your interest in adding trend lines to the scatter plot. However, we regret to inform you that this would not be appropriate in this case. The data points on the scatter plot represent independent food items, and as such, they are not part of a continuous dataset that would support a trend analysis. The scatter plot was specifically designed to illustrate how children perceive individual food items in terms of healthiness (healthy/unhealthy) and preference (like/dislike). Comment 9: Lines 509-511: The following are the observed deficiencies with reference to the report: Weak transition from results to the generalization of the results. In what ways does the study and the identified patterns connect to other nutritional problems in the Philippines? Response 9: We believe that this study makes a significant contribution to the understanding of children's nutrition in the Philippines. It provides valuable insights that can inform the development of nutrition education programs aimed at addressing malnutrition. For building strong transition and connection We include this in the text specifically in conclusion part: The study provides valuable insights that can inform the development of nutrition education programs aimed at addressing malnutrition Page 19 Line 686-688 Thank you for pointing this out, this is a very important point of view: In the manuscript, we pointed out: · Children face challenges in identifying "Grow" and "Glow" food items, as revealed by the ranked analysis of the Food Knowledge questionnaires. This difficulty highlights the need for early education, as we indicated that "At an early age, Filipinos should correctly recognize Grow and Glow food items due to their crucial role in growth, development, and the prevention of deficiencies and diet-related diseases in adulthood." This rationale has been incorporated into the manuscript. · Additionally, the children's drawings further indicate a low preference for fruits and vegetables, as evidenced by the limited number of drawings. We believe this reflects existing micronutrient deficiencies and underscores the importance of early intervention to improve children's understanding and preference for nutritious foods, which are essential in addressing malnutrition and promoting healthier diets with special emphasis in fruits and vegetable · This rationale has been incorporated into the manuscript. “Grow foods” are nutrient-dense foods rich in protein; examples are meat, poultry, eggs, milk, cheese, beans, and lentils. Protein-rich food helps our body to grow, build, and repair muscles and tissue and strengthen our bones and teeth. Research studies established that the proper amount of intake of protein is closely related to growth and development; therefore, consuming foods that are rich in protein facilitates catch-up growth in stunted children[36]. In the Philippines, according to the 2019 Expanded National Nutrition Survey (ENNS), stunting remains a moderate public health concern, recording that one (1) in four (4) Filipino school-aged children was stunted (24.9%). On the other hand, “Glow foods” are nutrient-dense foods rich in micronutrients such as vitamins and minerals, dietary fiber, antioxidants, and phytochemicals, which help regulate bodily processes. All fruits and vegetables belong to this group classification. Numerous studies have established that the consumption of fruits and vegetables is an essential component of a children`s balanced diet and can reduce the risk of experiencing chronic diseases later in life [37]. In the Philippines, micronutrient deficiency across age groups remains a public health concern. This includes deficiencies in Vitamin A, iodine, zinc, and iron, which can lead to the hidden hungers of malnutrition such as Vitamin A deficiency, Iodine deficiency disorder, zinc deficiency, and Iron deficiency anemia [38]. Micronutrient deficiencies can cause both morbidity and mortality. It can lead to serious consequences such as stunting, cognitive impairment, susceptibility to infection, birth defects, and lower school performance[39]. At an early age, Filipinos should correctly recognize Grow and Glow food items because of their significant role in the growth, development, and prevention of deficiency and other diet-related diseases in later life. This can be seen on Page 14, Line 444-466 · The findings of the study could serve as a valuable guide in designing nutrition education programs that incorporate creative approaches. These methods should aim to enhance, rather than limit, children's understanding of nutrition. By using engaging and interactive strategies, such as drawing, games, and storytelling, children can develop a deeper and more enjoyable connection with healthy eating habits. This could help overcome challenges like low preference for fruits and vegetables, and foster long-term positive changes in dietary behaviors, ultimately contributing to better nutrition and overall health. Comment 10: Line 634-638: A great strength is that the conclusion effectively sums up the findings made in the course of the analysis, while the final message is rather weak. Could you insert a more engaging phrase which could serve as an umbrella to the more specific findings of the research in terms of the public health implications? Response 10: Thank you for highlighting this: We added this to the manuscript: These insights could ultimately inform more targeted and effective public health strat-egies, exploring children’s food preferences through drawing creates age-appropriate and creative methods. Incorporate characters or foods from their drawings to make campaigns relatable creating visually engaging materials for nutrition education campaigns in the public health setting. This can be seen on Page 14, Line 444-466 Thank you so much for your time and effort for reviewing our paper, we have a valuable insight at your comments that is not only valuable for this paper for also in our future endeavors |
||||||||||||||||||||||||||||||||||||||||||||||||||||||||||||
